# RANDOM FEATURE MEAN-SHIFT

## ABSTRACT

Locating the modes of a probability density function is a fundamental problem in many areas of machine learning. However, classical mode-seeking algorithms such as mean-shift and its variants exhibit quadratic complexity with respect to the number of data points due to exhaustive pairwise kernel computation - a well-known bottleneck that severely restricts the applicability. In this paper, we propose **Random Feature mean-shift (RFMS)**, a novel linear complexity mode-seeking algorithm. We give a sampling-based estimator using random feature kernel approximation and zeroth-order gradient method that allows us to provably achieve linear runtime per iteration, with comprehensive theoretical guarantees for mode estimation and convergence behavior. Empirical evaluations on clustering and pixel-level image segmentation tasks show RFMS is up to 12x faster when compared with other mean-shift variants, offering substantial efficiency gains while producing near-optimal results. Overall, RFMS offers a practical and principled framework for scalable mode-seeking beyond kernel-value approximation, with explicit guarantees on the induced mode landscape and optimization dynamics.

## 1 INTRODUCTION

Mode-seeking refers to the identification of maxima ("modes") of a probability density. It provides a principled way to summarize and organize complex multimodal distributed data. This technique is routinely used in a variety of unsupervised learning tasks, notably clustering [13; 32], image processing [15; 38], and object tracking [16; 74]. For decades, Mean-shift [20] and its many variants have been the de facto algorithm for mode-seeking, the algorithm iteratively shifts each data point towards regions of higher density by computing local means within a predefined window. As a non-parametric method, mean-shift requires minimal assumptions about data distributions, making it highly versatile and effective.

The idea of mean-shift is closely related to kernel density estimation (KDE), where the gradients of the KDE function are employed to guide the points toward local maxima, representing the modes of the underlying probability density function. Figure 1 demonstrates the simple geometric intuition behind mean-shift. One natural way of interpreting the mean-shift procedure is gradient ascent,

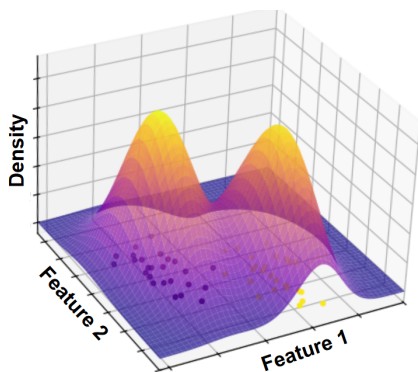

Figure 1: Kernel density estimation over synthetic 2D data(with 2 clusters) using Gaussian kernel. Points converge to the modes of empirical distribution in mean-shift.

in the sense that the point will eventually converge to the modes of KDE. Despite its effectiveness, conventional mean-shift methods face significant computational limitations, primarily stemming from their quadratic complexity related to exhaustive pairwise kernel function computation required for KDE. This bottleneck restricts the applicability of mean-shift to large-scale datasets and computationally intensive applications. Existing works on efficient mean-shift algorithms have explored solutions such as discretizing the feature space via hashing [38], employing specialized data structures for distance-based queries [72], or accelerating the algorithm in hardware [35]. Despite those efforts, none have succeeded in addressing the complexity bottleneck of mean-shift asymptotically without discretizing data representation in some fashion. Noticing the close connection between kernel density

estimation using shift-invariant kernels [1] and random Fourier feature kernel approximation [65; 21]. In this paper, we propose a novel sampling-based estimator for the mean-shift process. Based on random feature kernel approximation and zeroth-order optimization, RFMS is useful in many areas of machine learning. Overall, our contributions are as follows:

- **Algorithm**: We propose Random Feature mean-shift (RFMS), an asymptotic complexity improvement of the standard mean-shift algorithm(from $\mathcal{O}(n^2)$ to $\mathcal{O}(n)$ with respect to the number of data points), enabling efficient density-based analysis and mode-seeking.
- **Theory**: Strong and comprehensive theoretical guarantees for RFMS are provided. We first establish high-probability concentration bounds for random feature KDE approximation (Theorem 1, 2). Furthermore, error bounds for mode estimation and tracking are also provided(Theorem 5, 6), ensuring the results obtained by RFMS are close to the actual modes of kernel density estimation.
- **Evaluation**: Experiments against other mean-shift variants on clustering and pixel-level image segmentation demonstrate RFMS's advantage on reducing computational cost (up to 12x speedup on clustering and 3x on segmentation) while providing nearly optimal results.

Our work saliently and naturally brings together density estimation, random feature approximation, and mode-seeking into a single end-to-end framework. To the best of our knowledge, this is the first work to provide high-probability random feature-induced perturbation guarantees for KDE, and propagate these approximation effects into principled mode-stability and convergence guarantees for an efficient mode-seeking algorithm.

## 2 RELATED WORKS

**Mean-shift:** The mean-shift algorithm has been extensively studied as a powerful non-parametric technique for locating the modes of a density function. The algorithmic basis of mean-shift was first introduced by Fukunaga and Hostetler [20], who proposed a method to estimate the gradient of a multivariate probability density function and perform mode-seeking iteratively. This initial formulation was later improved and popularized through the lens of clustering [13] and computer vision tasks such as edge-preserving smoothing and image segmentation [15]. The mean-shift algorithm is computationally expensive. Specifically, one iteration of mean-shift on a single point requires computing the kernel values between all other points, resulting in a computational complexity of $\mathcal{O}(n^2)$ per iteration, where $n$ is the number of data points. Modern implementations of mean-shift algorithms typically utilize a flat kernel and efficient data structures, such as ball-tree or kd-tree, to organize distance searches over data points [62; 72]. This, however, does not improve the asymptotic complexity. More recently, mean-shift++[39] by Jang et al. proposes to hash data points into discrete hypercubes at the beginning of every iteration, allowing mean-shift to be performed by averaging the neighboring hypercubes instead of all other points. However, as the number of hypercubes scales exponentially with data dimensionality, MS++ does not scale well into higher dimensions. Despite mean-shift being the de facto and the most used mode-seeking method in practice, we note that other methods for finding mode of a KDE exist, notably the works by Lee et al. [54] and Luo et al. [54]. However, they primarily aim to find an approximately global maximizer of the KDE with dimensionality reduction plus additional heavy algorithmic machinery (e.g., solving polynomial systems). In contrast, RFMS is designed to run a mean-shift–style mode-seeking process for many modes, which is necessary in clustering and segmentation. Furthermore, RFMS is designed to use only simple and fully vectorized computation (essentially just dot products and some simple element-wise operations which emphasizes scalability and efficiency.

**Random Features:** Emerging from the literature on kernel methods, Random Features(RFF) [65] is a strategy to scale up kernel-based learning algorithms. The central idea behind random feature methods is to approximate a shift-invariant kernel $k(x, x') = k(x - x')$ (i.e. Gaussian, Laplacian and Cauchy kernels) using an explicit randomized feature map $\phi : \mathbb{R}^d \to \mathbb{R}^D$ or $\mathbb{C}^D$ such that $k(x, x') = \mathbb{E}[\langle \phi(x), \phi(x') \rangle]$, the expectation is taken with respect to the construction of $\phi$. This is theoretically grounded in Bochner's theorem, which states that any continuous, shift-invariant, positive definite kernel $k$ on $\mathbb{R}^d$ is the Fourier transform of a probability measure. Specifically, let $p(\omega)$ be the spectral density of the kernel, since $k(x - x') = \int_{\mathbb{R}^d} p(\omega)e^{j\omega \cdot (x - x')}d\omega$ , one can approximate the integral via Monte Carlo method [57] by sampling on frequencies $\omega$. The error of Random Feature approximation can be controlled by increasing $D$, the number of features. One

central challenge of Random Feature is to achieve a high-quality approximation with $D$ being as small as possible. Several variants and extensions have been developed, such as orthogonal random features [78] and Quasi Monte-Carlo [36] to reduce the size of $D$. Prior random feature works largely focus on approximating kernel values or improving the uniform-error bound behavior. In contrast, our work here provides an end-to-end framework for RFF-based mode-seeking, and we give theoretical insights on how random feature approximation induced errors can propagate into KDE approximation and mode stability.

**Zeroth-Order Optimization:** Also known as derivative-free optimization or black-box optimization, is a class of optimization methods that do not require first-order information (i.e., without computing gradients). Such techniques are often used for problems where gradient information is unavailable, expensive, or unreliable [51]. The central challenge in Zeroth-Order Optimization is approximating the gradient using only function values. One of the most common approaches is the finite-difference gradient estimator [58; 17], which estimates the gradient $\nabla f(x)$ by querying the function at perturbed points around $x$ via $\hat{\nabla} f(x) = \frac{f(x+\mu u)-f(x)}{\mu} u$, where $u$ is a random direction and $\mu$ is a small scalar called smoothing parameter. Particularly, if $u$ is sampled from multivariate Gaussian or uniformly from a sphere of radius the same as the dimension of the problem [51], then $\mathbb{E}_{u \sim \mathcal{N}(0, \mathbf{I}_d) \text{ or } \mathcal{U}(\mathcal{S}(0,d))} \left[ \hat{\nabla} f(x) \right] = \nabla f_\mu(x)$, which provides an unbiased estimation to the gradient of $f_\mu$, the smoothed version of $f$, useful for Stochastic Gradient methods [26; 52].

## 3 NOTATION AND PRELIMINARIES

Consider $\mathcal{D} = \{x_1, x_2, \ldots, x_n\} \subset \mathbb{R}^d$ a dataset of $n$ points in $d$-dimensional euclidean space. One way to model the probability density function of the underlying data generative process is to construct a kernel density estimation (KDE) [61] over observed data points: $\hat{f}(x) = \frac{1}{nc} \sum_{i=1}^n k(x, x_i) \, \forall x_i \in \mathcal{D}$. Here $k : \mathbb{R}^d \to \mathbb{R}_+$ :is a radially symmetric kernel function, and $c$ is some kernel dependent constant that ensures $\int_{\mathbb{R}^d} \hat{f}(x) dx = 1$ so it is a valid probability distribution. KDE represents a smoothed version of the empirical distribution, with the kernel function distributing mass around each point $x_i$. Although any shift-invariant kernel can be used in RFMS, for consistency and ease of notation, in this paper we consider $k$ to be the Gaussian kernel and denote $k_h$ the Gaussian kernel of bandwidth $h$: $k_h(x_i, x_j) = \exp\left(-\frac{1}{2h^2} \|x_i - x_j\|^2\right) = k_h(\Delta)$ where $\Delta = \|x_i - x_j\|^2$, and $\hat{f}_{k_h}$ is the KDE using $k_h$. As such $c = \int_{\mathbb{R}^d} k_h(\Delta) d\Delta = \left(2\pi h^2\right)^{d/2}$. One natural way of finding the modes of the KDE is to allow points to "climb" over the KDE via gradient ascent [20; 32]: $\nabla \hat{f}_{k_h}(x) = \frac{1}{h^2 cn} \sum_{i=1}^n k_h(x - x_i) \cdot (x_i - x)$. This expression shows that the gradient of the KDE points in the direction of a weighted average of the $(x_i - x) \forall x_i \in \mathcal{D}$. Notice that $\nabla \hat{f}_{k_h}$ is proportional to $m(x) = \frac{\sum_{i=1}^n x_i k_h(x - x_i)}{\hat{f}_{k_h}(x)} - x$ by a multiplicative factor of $\left(h^2 cn \hat{f}_{k_h}(x)\right)^{-1}$. Consider the gradient ascent $x^{(l+1)} = x^{(l)} + m(x^{(l)})$ which gives the modern fix-point iteration version of the mean-shift [15]: $x^{(l+1)} = \frac{\sum_{i=1}^n x_i k_h(x^{(l)} - x_i)}{\sum_{i=1}^n k_h(x^{(l)} - x_i)}$. In this way, since $x^{(l+1)} - x^{(l)} = m(x^{(l)}) = \left(h^2 cn \hat{f}_{k_h}(x^{(l)})\right)^{-1} \nabla \hat{f}_h(x^{(l)})$ we can also view the update rule as normalized gradient ascent (by the density at $x^{(l)}$) on the KDE. This allows us to define the mean-shift problem we aim to solve in this paper:

$$x_i^{(l+1)} = x_i^{(l)} + \eta \frac{\nabla \hat{f}_{k_h}(x_i^{(l)})}{\hat{f}_{k_h}(x_i^{(l)})} \quad \forall i \in \{1 \ldots n\} \quad \text{with} \quad x_i^{(0)} = x_i \quad \forall x_i \in \mathcal{D} \tag{1}$$

Similar mean-shift formulations are also studied in previous papers [1; 32; 20]. The main overhead of this procedure comes from the density estimation via KDE (denominator) and the gradient computation (numerator), both of which takes $\mathcal{O}(n^2)$ since each iteration of update requires computing pairwise kernel value between $x_i^{(l)}$ and $x_i \in \mathcal{D}$, which is the main bottleneck. It is worth noting that although many variants of the mean-shift algorithm have been proposed in existing literature, such as Blurring mean-shift [15; 7], Robust mean-shift [5], they all suffer from the same quadratic complexity

limitation. In this work, we demonstrate that it is in fact possible to construct an efficiency estimator for $\frac{\nabla \hat{f}_{k_h}(x_i^{(l)})}{\hat{f}_{k_h}(x_i^{(l)})}$ that takes constant time per point to achieve $\mathcal{O}(n)$ complexity per iteration of update.

# 4 RANDOM FEATURE MEAN-SHIFT (RFMS)

Here, we provide algorithmic details of Random Feature mean-shift (RFMS), a novel linear complexity mode-seeking algorithm. We first establish a framework for efficient kernel density estimation using random feature method and then utilize it for zeroth-order gradient ascent for mode-seeking.

**Random Feature Density Estimation:** As discussed in Section 3, one of the main bottleneck of mean-shift algorithm is the computation of local density as $\hat{f}_{k_h}(x)$ for any $x \in \mathbb{R}^d$ requires the pairwise kernel computation between $x$ and every $x_i$ in dataset $\mathcal{D}$. This becomes problematic if $\mathcal{D}$ is large. However, recall that $\hat{f}_{k_h}(x) = \frac{1}{nc} \sum_{i=1}^{n} k_h(x, x_i)$, which is the sum of kernel values with a fixed data point. Now consider a Random Feature [65] transformation for $k_h$ denoted as $\phi_{k_h} : \mathbb{R}^d \to \mathbb{C}^D$ with the property that $\langle \phi_{k_h}(x), \phi_{k_h}(x') \rangle \approx k_h(x, x') \forall x, x' \in \mathbb{R}^d$ for some sufficiently large $D$. We can then rewrite $\hat{f}_{k_h}(x)$:

$$\hat{f}_{k_h}(x) = \frac{1}{nc} \sum_{i=1}^{n} k_h(x, x_i) \approx \frac{1}{nc} \sum_{i=1}^{n} \langle \phi_{k_h}(x), \phi_{k_h}(x_i) \rangle = \frac{1}{nc} \left\langle \phi_{k_h}(x), \sum_{i=1}^{n} \phi_{k_h}(x_i) \right\rangle \quad (2)$$

The above derivation uses the fact that the complex inner product is linear in the second argument. We can then define $\Phi = \frac{1}{nc} \sum_{i=1}^{n} \phi_{k_h}(x_i)$, and density estimation can be done by $\hat{f}_{k_h}(x) \approx \langle \phi_{k_h}(x), \Phi \rangle$. As a reminder $c = (2\pi h^2)^{d/2}$ for Gaussian kernel of bandwidth $h$ over $\mathbb{R}^d$ and complex vector inner product $\langle a, b \rangle = (\overline{a} \cdot b) \forall a, b \in \mathbb{C}^D$. To construct $\phi_{k_h}$, simply sample random features $\{\omega_1 \ldots \omega_D\}$ from $\omega \sim \mathcal{N}\left(0, \frac{1}{h^2} \mathbf{I}_d\right)$ and we have:

$$\phi_{k_h}(x) = \frac{1}{\sqrt{D}} \left[ e^{j(\omega_1^T x)} e^{j(\omega_2^T x)} e^{j(\omega_3^T x)} \ldots e^{j(\omega_D^T x)} \right] \quad (3)$$

Here, $j$ is the imaginary unit. Let $\hat{g}(x) = \frac{1}{nc} \langle \phi_{k_h}(x), \Phi \rangle$, it is an unbiased estimation of the KDE:

$$\hat{f}_{k_h}(x) = \mathbb{E}_{\omega \sim \mathcal{N}\left(0, \frac{1}{h^2} \mathbf{I}_d\right)} [\hat{g}(x)] \quad \forall x \in \mathbb{R}^d \quad \text{(See Appendix E for the proof).} \quad (4)$$

We will later give a high-probability bound on the concentrations around this expected value. One important caveat is that although the expectation of $\hat{g}(x)$ is real, in an approximation setting where $D$ is finite, the value of $\hat{g}(x)$ is not guaranteed to be real-valued. However, since KDE is a real-valued function, it implies that the imaginary part has zero mean therefore adds only variance, so in practice we only take the real part: $\text{Re}(\hat{g}(x)) = \text{Re}(\langle \phi_{k_h}(x), \Phi \rangle) = \frac{1}{nc} \sum_{i=1}^{n} \text{Re}(\langle \phi_{k_h}(x), \phi_{k_h}(x_i) \rangle)$. What this means is that when evaluating the KDE at some point in $\mathbb{R}^d$ with respect to a dataset $\mathcal{D}$, instead of compute $n$ kernel values, one can get an approximation in constant time via $\hat{g}(x)$ because the mapping $\phi_{k_h}(x)$ takes constant time per point and $\Phi$ only needs to be computed once over $\mathcal{D}$ at the beginning and stored.

**Zeroth-Order Gradient Estimation:** Recall in Equation 1, in order to construct the mean-shift updates for a point $x \in \mathbb{R}^d$ using a dataset $\mathcal{D} \subset \mathbb{R}^d$, it is necessary to obtain the ascending-direction vector $\nabla \hat{f}_{k_h}(x)$, which is the average of $(x_i - x) \forall x_i \in \mathcal{D}$ weighted by $k_h(x, x_i)$. Previously, we have established that one can estimate the KDE value $f_{k_h}(x)$ by querying $\hat{g}(x) = \langle \phi_{k_h}(x), \Phi \rangle$. Naviely, one could take the gradient of $\hat{g}(x)$ as a surrogate to the actual KDE gradient; however, that would require computing the Jacobian w.r.t every dimension of $x$, which will result in an complexity of $\mathcal{O}(Dd)$ per point per iteration. To further reduced the mean-shfift cost to $\mathcal{O}(D)$ per point per iteration, we apply zeroth-order gradient method instead of calculating the gradient analytically. Therefore, the ability to query density estimation is powerful as it enables us to estimate $\nabla \hat{f}_{k_h}(x)$ via a zeroth-order gradient estimator, particularly the 2-point forward estimator taking the form:

$$\hat{\nabla} \hat{f}_{k_h}(x) = \frac{\text{Re}(\hat{g}(x + \mu u) - \hat{g}(x))}{\mu} u \quad (5)$$

Here we sample $u$, the random ascent direction, uniformly from the standard Gaussian $u \sim \mathcal{N}(0, \mathbf{I}_d)$. And $\mu$ is a small smoothing parameter set beforehand. This estimator works by probing the function

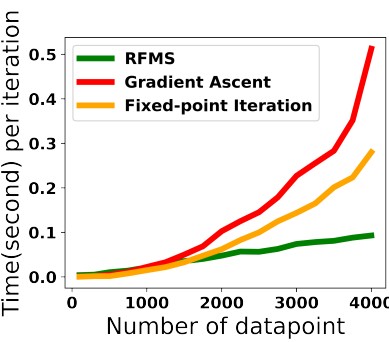
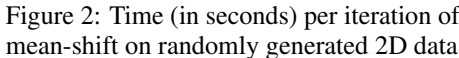

Figure 2: Time (in seconds) per iteration of mean-shift on randomly generated 2D data.

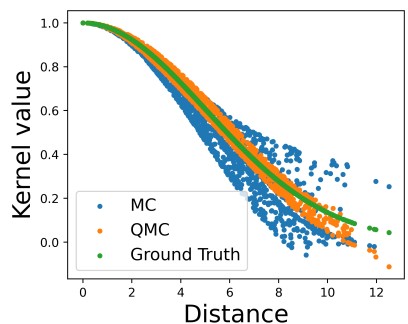

Figure 3: Kernel approximation quality using MC and QMC, $D = 30$.

in a randomly chosen direction and measuring the change in function output after taking a small step. This change is used to estimate the directional derivative. Geometrically, it captures how steeply the function rises in that sampled direction, giving a noisy ascent direction. Repeating this across iterations allows a point to follow the landscape of the function without ever computing actual gradients. Two levels of approximation are happening in Equation5, the first one being that $\hat{g}(x)$ itself is an unbiased estimation of the KDE $\hat{f}_{k_h}(x)$. The second one is the gradient estimation of the KDE, namely the $\hat{\nabla}\hat{f}_{k_h}$, which is unbiased with respect to the gradient of a smoothed version of $\hat{f}_{k_h}$. With that, we can arrive at a new update rule for mean-shifting:

$$x_i^{(l+1)} = x_i^{(l)} + \eta \frac{\hat{\nabla}\hat{f}_{k_h}(x_i^{(l)})}{\mathrm{Re}\left(\hat{g}(x_i^{(l)})\right)} \quad \forall i \in \{1 \dots n\} \quad \text{with} \quad x_i^{(0)} = x_i \quad \forall x_i \in \mathcal{D} \tag{6}$$

Here, $\eta$ is the learning rate, since the core characteristic of zeroth order optimization is that it only requires evaluating function values (in the context of KDE, the $\hat{f}_{k_h}$), it is worth noting that other, potentially more advanced, zeroth order gradient estimator [50; 12] can be used as drop-in replacement in RFMS which can lead to further variance reduction in gradient estimation and faster convergence.

**Complexity & Scalability:** Essentially, RFMS estimates the mean-shift update via two density estimations; as such, by using $\mathrm{Re}(\hat{g})$, the cost of RFMS per iteration becomes constant per data point and scales linearly in the size of $\mathcal{D}$. One other cost of RFMS comes from the number of random features used in $\phi_{k_h}$, making the final time complexity $\mathcal{O}(nD)$ for RFMS in contrast to the $\mathcal{O}(n^2 d)$ complexity of standard mean-shift algorithm (both in the gradient ascent form and fixed-point iteration form). In situations where the dataset is very small, since $D$ is typically larger than $d$, standard mean-shift can potentially be faster than RFMS. However, in practical settings, RFMS is much more suited for scaling to larger tasks. In Fig. 2, we empirically demonstrate the scalability of RFMS on randomly generated 2D data, showing a minimal increase in cost (time per iteration) when compared to standard mean-shift algorithms. RFMS inherits the standard RFF trade-off between efficiency and approximation quality as $D$ increases, but this approximation is widely accepted because it replaces quadratic kernel computations with simple feature inner products and yields major scalability gains in large-$n$ settings.

## 5 ESTIMATION AND CONVERGENCE BOUNDS

$\mathrm{Re}(\hat{g})$ serves as a surrogate of the actual KDE function $\hat{f}_{k_h}$. The quality of RFMS largely depends on how well it approximates KDE. In this section, we first establish an error bound on estimating $\hat{f}_{k_h}$ with $\mathrm{Re}(\hat{g})$. Then, we show conditions under which the modes of the $\hat{f}_{k_h}$ and $\mathrm{Re}(\hat{g})$ are close and bounded. Lastly, utilizing existing results regarding zeroth-order optimization, we show the convergence of RFMS to points near the modes of $\hat{f}_{k_h}$.

**Error bound on estimating $\hat{f}_{k_h}(x)$ with $\hat{g}(x)$:** The goal here is to give a bound on the error of estimating KDE value with $\hat{g}$. That is, we want to bound $\left|\hat{g}(x) - \hat{f}_{k_h}(x)\right|$. By applying Hoeffding's inequality, we can derive the following bound:

**Theorem 1** *For any point $x \in \mathbb{R}^d$, chose any $\delta \in (0,1)$, with probability at least $1 - \delta$:*

$$\left| \hat{g}(x) - \hat{f}_{k_h}(x) \right| \leq \frac{4}{c} \sqrt{\frac{1}{2D} \ln \frac{4}{\delta}} \quad \text{(See Appendix F for the proof.)} \tag{7}$$

Furthermore, since the $\hat{g}(x)$ concentrates around a real number, the complex part of $\hat{g}(x)$ contribute only variance, so if KDE is approximated via $\mathrm{Re}\,(\hat{g}(x))$, the bound can be further reduced to $\frac{2}{c} \sqrt{\frac{1}{2D} \ln \frac{4}{\delta}}$. For the remainder of this section, we will only consider the real part of $\hat{g}(x)$. Extending classicial results on random feature method [65], we then extend Theorem1 to a uniform convergence bound:

**Theorem 2** *Let $\mathcal{X}$ be a compact set over $\mathbb{R}^d$ such that $\mathcal{D} \subset \mathcal{X}$. Denote $\mathrm{diam}(\mathcal{X})$ the diameter of $\mathcal{X}$. Then, for error tolerance $\epsilon$, the following bound holds:*

$$\mathrm{Pr}\left( \sup_{x \subset \mathcal{X}} \left| \mathrm{Re}\,(\hat{g}(x)) - \hat{f}_{k_h}(x) \right| \geq \epsilon \right) \leq 2^8 \left( \frac{c\sqrt{d}\,\mathrm{diam}(\mathcal{X})}{h\epsilon} \right)^2 \exp\left( -\frac{D\epsilon^2}{c^2 4(d+2)} \right) \tag{8}$$

*(See Appendix G for the proof.)*

The above yields a uniform additive bound for approximating KDE via RFF. The primary reason for providing an additive error bound instead of a relative error bound is that the density can be arbitrarily close to $0$ in low-density regions, hence a relative error bound over the entire space is generally ill-posed unless one restricts to regions with meaningful density values [10; 43]. If we pose constrain to a subset $S \in \mathbb{R}^d$ such that $S = \left\{ x : \hat{f}_{k_h}(x) \geq \tau \right\}$, our uniform additive error bound immediately implies a relative bound: $\sup_{x \in S} \left| \left( \mathrm{Re}\,(\hat{g}(x)) - \hat{f}_{k_h}(x) \right) / \hat{f}_{k_h}(x) \right| \geq \frac{\epsilon}{\tau}$.

**Mode Stabillity:** In RFMS, we use $\mathrm{Re}\,(\hat{g})$ as an surrogate of the KDE $\hat{f}_{k_h}$. Therefore, for mode-seeking purposes, we would like to show that the modes of $\hat{f}_{k_h}$ and $\mathrm{Re}\,(\hat{g})$ are close. To achieve this, we first demonstrate the point-wise closeness of the gradient and the Hessian.

**Theorem 3** *$C$ is a universal constant, for any point $x \in \mathbb{R}^d$, chose any $\delta \in (0,1)$, with probability at least $1 - \delta$:*

$$\left\| \nabla \mathrm{Re}\,(\hat{g}(x)) - \nabla \hat{f}_{k_h}(x) \right\| \leq \frac{1}{nc} \left( \frac{8Cen\sqrt{d}\ln(2/\delta)}{Dh} + \sqrt{\frac{8Cen^2 d \ln(2/\delta)}{Dh^2}} \right) \tag{9}$$

*(See Appendix H for the proof.)*

**Theorem 4** *For any point $x \in \mathbb{R}^d$, chose any $\delta \in (0,1)$, with probability at least $1 - \delta$:*

$$\left\| \nabla^2 \mathrm{Re}\,(\hat{g}(x)) - \nabla^2 \hat{f}_{k_h}(x) \right\|_F \leq \frac{1}{nc} \left( \frac{8eCnd\ln(2/\delta)}{Dh^2} + \sqrt{\frac{8eCn^2 d\sqrt{d(d+2)}\ln(2/\delta)}{Dh^4}} \right) \tag{10}$$

*(See Appendix I for the proof.)*

Then, assume $\hat{f}_{k_h}$ is a Morse function (a function with non-degenerate critical points). Denote $\mathrm{Lip}(\cdot)$ the Lipschitz constant of a function, and $\lambda_{min}(\cdot)$ the smallest eigenvalue of a square matrix. We can then establish the conditions regarding the closeness between their critical points (modes):

**Theorem 5** *Let $x^*$ be any critical point of $\hat{f}_{k_h}$, define: $\alpha = \frac{\mathrm{Lip}(\nabla^2 \mathrm{Re}(\hat{g}))\epsilon_1}{\left(\lambda_{min}(\nabla^2 \hat{f}_{k_h}(x^*)) - \epsilon_2\right)^2}$. If : (1). Chose $\epsilon_2 \leq \lambda_{min}\left(\nabla^2 \hat{f}_{k_h}(x^*)\right)$. (2). Chose $\epsilon_1$ such that $\alpha \leq \frac{1}{2}$. (3). $\left\| \nabla \mathrm{Re}\,(\hat{g}(x)) - \nabla \hat{f}_{k_h}(x) \right\| \leq \epsilon_1$ and $\left\| \nabla^2 \mathrm{Re}\,(\hat{g}(x)) - \nabla^2 \hat{f}_{k_h}(x) \right\|_F \leq \epsilon_2$ both holds with probability at least $1 - \delta/2$. Then, with probability at least $1 - \delta$, there is only one critical point $\hat{x}^*$ of $\mathrm{Re}\,(\hat{g})$ such that:*

$$\hat{x}^* \in B\left( x^*, \frac{\epsilon_1}{\lambda_{min}\left(\nabla^2 \hat{f}_{k_h}(x^*)\right) - \epsilon_2} \right) \quad \text{(See Appendix J for the proof.)} \tag{11}$$

This result enables the quantification of how much each mode shifts between the KDE $\hat{f}_{k_h}$ and the random feature approximated KDE $\mathrm{Re}\,(\hat{g})$. Since we can make $\epsilon_1$ and $\epsilon_2$ arbitrarily small by increasing $D$ (see theorem 3 and theorem 4), with properly chosen $\epsilon_1$ and $\epsilon_2$, the mode change can be well controlled, hence making $\mathrm{Re}\,(\hat{g})$ a good surrogate for mode-seeking. Our gradient/Hessian and mode-closeness guarantees (Theorems 3–5) imply that the true modes are preserved and can shift only by a small amount under the random-feature approximation. Any additional modes, if they occur, must arise from minor oscillations of the approximated density in low-density regions; such minor oscillations are typically negligible for mode-seeking and do not affect the algorithm's behavior. To empirically substantiate these claims, Appendix B presents synthetic 1D and 2D experiments that visualize and compare the random-feature density approximation against the exact KDE surface. The results are consistent with the theory.

**Convergence of RFMS:** Recall the RFMS iteration in equation 6, the algorithm can be interpreted as running zeroth-order gradient ascent on $\mathrm{Re}\,(\hat{g})$ with decaying step-size. This interpretation allows us to analyze the convergence of RFMS with existing results from zeroth-order optimization literature.

**Theorem 6** *Suppose a point $x \in \mathbb{R}^d$ has a local mode $\hat{x}^*$ of $\mathrm{Re}\,(\hat{g})$ with Łojasiewicz exponent $\theta$. Let $l^* \in \{0 \cdots T\}$ be the iteration index such that $\mathbb{E}_{u \sim \mathcal{N}(0,\mathbf{I}_d)} \left\| \nabla \mathrm{Re}\,(\hat{g}\,(x^{(l)})) \right\|^2$ is the smallest. Then:*

$$\mathbb{E}_{u \sim \mathcal{N}(0,\mathbf{I}_d)} \left[ \left\| \hat{x}^* - x^{(l^*)} \right\|^2 \right] \leq \mathcal{O}\left( \left( \frac{1}{\sqrt{T}} + \mu^2 d^2 \right)^{1/2\theta} \right) \textit{(See Appendix K for the proof.)} \quad (12)$$

Combined with Theorem 5, the above result allows one to quantify how close the solution returned by RFMS is to an actual mode of the KDE. We believe this "kernel approximation + mode stability + zeroth-order optimization" synthesis is nontrivial and substantial, because it directly addresses what matters for mean-shift: not just approximating the kernel function, but preserving the mode structure that defines clusters.

# 6 Implementation Detail

The pseudocode for RFMS is presented in Alg. 1. It takes in a set of points in $\mathbb{R}^d$ and outputs the shifted version of those points also in $\mathbb{R}^d$. We provide additional details on RFMS to enhance its efficiency and extend it to applications that require a blurring process.

---

**Algorithm 1:** Random Feature mean-shift (RFMS)

**Data:** Dataset $\mathcal{D}$, bandwidth $h$, smoothing parameter $\mu$, Learning rate $\eta$, number of iteration $T$, RFF dimension $D$

1 $\phi_{k_h} \leftarrow$ generate random feature mapping for gaussian kernel with band width $h$ via QMC;
2 $e_i \leftarrow \phi_{k_h}(x_i) \quad \forall x_i \in \mathcal{D};$     /*Encode each data point into*/
3 $\Phi \leftarrow \frac{1}{nc} \sum_{i=1}^{n} e_i;$     /*Define $\Phi$ to be used in $\hat{g}(x)$*/
4 $x_i^{(0)} \leftarrow x_i \quad \forall x_i \in \mathcal{D};$     /*Initial position*/
5 **for** $l = 0, 1, 2, 3 \ldots T$ **do**
6     **for** *each $x_i^{(l)}$* **do**
7        $u \leftarrow \mathcal{N}(0, \mathbf{I}_d)$ ;     /*Sample random ascent direction*/
8        $x_i^{(l+1)} \leftarrow x_i^{(l)} + \eta \left( \left( \hat{\nabla} \hat{f}_{k_h}\left(x_i^{(l)}\right) \right) / \mathrm{Re}\left(\hat{g}\left(x_i^{(l)}\right)\right) \right)$ ;     /*Update position*/
9     **end**
10     **if** *Blurring* **then**
11        $\Phi \leftarrow \frac{1}{nc} \sum_{i=1}^{n} e_i^{(l+1)};$     /*Update $\Phi$ to reflect new density function after shifting*/
12     **end**
13 **end**
14 **return** $\{x_i^{(T)}\};$     /*New points after mean-shift*/

---

**Construct Random Feature Mapping using Quasi Monte-Carlo:** We use the random feature method in equation 3 for kernel approximation. As discussed previously, one core challenge here is

to reduce the number of features used while still providing a good approximation; in other words, how to make $D$ as small as possible while still providing a high-quality approximation. Since evaluating a shift-invariant kernel is essentially the same as evaluating an integral associated with the kernel(Bochner's Theorem [66]), one prominent solution is to incorporate Quasi Monte-Carlo (QMC) techniques for numerical integration into the random feature framework [36; 76]. QMC uses low-discrepancy sequences (e.g., Sobol, Halton, or Faure sequences) to generate random features that cover the space more uniformly. To put it simply, instead of sampling frequencies $\omega \sim \mathcal{N}\left(0, \frac{1}{h^2}\mathbf{I}_d\right)$, we can improve RFMS by sample $\omega$ from Halton sequence and apply inverse cumulative distribution function to move them into the correct distribution. The intuition behind QMC is that well-distributed deterministic sampling can outperform random sampling in integration and approximation tasks. We demonstrate the improvement of QMC In Fig. 3 where we generate random 2D points and compute pairwise kernel values with $k_h$, $\phi_{k_h}$ constructed with Monte-Carlo, and $\phi_{k_h}$ constructed use QMC. As shown, with the same $D$, QMC can produce much higher-quality approximations, especially when the actual kernel value is small. The use of QMC allows us to reduce $D$, hence further improving the computational efficiency of RFMS.

**Non-blurring vs. Blurring mean-shift:** In non-blurring setting, each point climbing a hill (mode) based on the fixed landscape: $x^{(l+1)} = \frac{\sum_{i=1}^{n} x_i k_h(x^{(l)}-x_i)}{\sum_{i=1}^{n} k_h(x^{(l)}-x_i)}$. The landscape is fixed in the sense that kernels are computed with unshifted points. However, in many application scenarios such as image smoothing [15], data consolidation [7], or structure-preserving denoising [34], blurring mean-shift: $x^{(l+1)} = \frac{\sum_{i=1}^{n} x_i k_h(x^{(l)}-x_i^{(l)})}{\sum_{i=1}^{n} k_h(x^{(l)}-x_i^{(l)})}$ is preferred for its faster convergence due to data contraction. This blurring process can be easily integrated into RFMS. Simply view the blurring as gradient ascent over a new KDE based on shifted points at every iteration. We can update $\Phi$ at the end of every iteration, so the $\text{Re}(\hat{g}(x))$ would produce an estimated KDE value over shifted points.

**Representing shift via element-wise multiplication:** Alg. 1 requires going back and forth between $\mathbb{R}^d$ and $\mathbb{C}^d$. However, with the help of complex number properties, it is possible to run RFMS entirely on the encoded version of the data points. Consider $\phi_{k_h}$ the encoding function, we can then represent translation(shift) in $\mathbb{R}^d$ via element-wise multiplication in $\mathbb{C}^d$ based on the property that:

$$\phi_{k_h}(x + x') = \frac{D}{\sqrt{D}}\phi_{k_h}(x) \otimes \phi_{k_h}(x') \quad \forall x, x' \in \mathbb{R}^d \quad \text{(See Appendix L for the proof).} \tag{13}$$

Where $\otimes$ denotes element-wise multiplication. This is the primary reason we chose to use the complex version of the random feature instead of the real-valued version, as the real-valued version is unable to achieve the same results due to the periodic nature of the cosine function. In this way, the original data can be discarded after the encoding, and subsequent operations can be performed exclusively on the encoded version of the data:

$$e_i^{(l+1)} = \frac{D}{\sqrt{D}}e_i^{(l)} \otimes \phi_{k_h}\left(\eta \frac{\hat{\nabla}\hat{f}_{k_h}(x^{(l)})}{\text{Re}\left(\hat{g}(x^{(l)})\right)}\right) \tag{14}$$

The capability of updating encodings in place is appealing as it further simplifies the algorithm. Although instead of shifted points, the algorithm will give $\{e_i^{(T)}\}$ with pairwise inner products approximating $k_h$ over shifted points. Since there exists a one-to-one correspondence between kernel value and distance, this is sufficient for any subsequent kernel or distance-based algorithms.

# 7 EXPERIMENTS

We verify the effectiveness and applicability of RFMS, we first directly inspect the mode-seeking behavior of RFMS using randomly synthesized clusters of different variance and cluster shape. The data points at different iterations are visualized and shown in Figure 4. As demonstrated, RFMS indeed achieves the intended mode-seeking functionality. It is also worth noting that the observations here closely match the theoretical insights we provided in Section 5. With a relatively small $D = 200$, we observe the points converge to a point very close to the actual modes (Theorem 5). Since directly evaluating mode-seeking algorithms is difficult, we instead apply RFMS in two applications where mean-shift algorithms are often applied - **(1).** In section 7.1, the RFMS algorithm is evaluated against other mean-shift algorithms in the context of clustering. We report clustering quality and time consumption of different methods. **(2).** In section 7.2, we apply RFMS to pixel-level image segmentation, a practical area of interest in computer vision. We use QMC for both experiments.

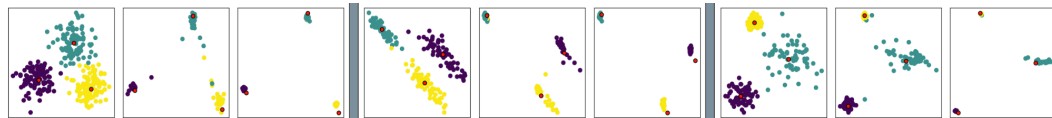

Figure 4: Trajectory Visualization of RFMS at $T = \{0, 30, 100\}$ and $D = 200$ on three separate examples with varying variance and cluster shape. Red points mark actual modes of clusters.

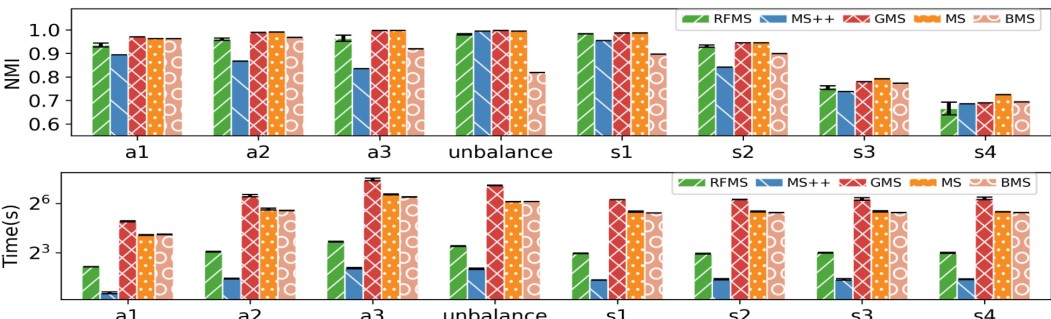

Figure 5: Comparison of different mean-shift algorithms in terms of time (measured in seconds in log scale) and normalized mutual information(NMI).

### 7.1 MEAN-SHIFT CLUSTERING

Mean-Shift algorithms are routinely used as clustering algorithms. It is particularly useful as it does not require the number of clusters to be predefined and can discover arbitrarily shaped clusters, given cluster forms a density peak. RMFS, as an efficient approximator of the classical mean-shift formulation, can also be used in this way. Here, we compare RFMS against other types of mean-shift algorithms: **(MS).** Fixed point iteration [9; 47; 3; 5]. **(GMS).** Gradient ascent over KDE [32; 20]. **(BMS).** Blurring mean-shift [9; 28; 71]. **(MS++).** Grid-based hashing [38]. Points that converge to the same mode are considered a cluster. After applying mean-shift, the connected component algorithm [8] is used to assign data points into different clusters. We conduct experiments on eight clustering benchmarking datasets from Fänti et al. [18]. We use $T = 100$ across all methods and $D = 300$ for RFMS. Each experiment was run 5 times, and we report the mean and standard deviation. The main clustering results are presented in Figure 5. Another huge advantage RFMS has is that its computational complexity scales well with $d$, the dimension of the data. The same is not true for hashing-based methods like MS++. To verify this, we conduct additional experiments on higher-dimensional real datasets [19; 40; 4]. The NMI and time results are shown in Table 1.

**Results:** We observe that RFMS can produce nearly optimal clustering results (on average 0.03 NMI drop-off compared with best NMI on each dataset) while being significantly more efficient than conventional mean-shift algorithms (MS, BMS, and GMS), with up to a 12x speed-up. We also observe that the efficiency benefit becomes more significant as the dataset grows larger, which is due to the asymptotic complexity improvement of RFMS, thereby verifying its scalability. Despite RFMS being slightly slower than MS++ (On average, 5.6s slower), RFMS produced better clustering quality in 6 out of 8 datasets tested in terms of normalized mutual information. In Table 1, on additional datasets, with increasing ambient dimension $d$, the efficiency of MS++ drops significantly. Particularly, when $d = 7$(WirelessLocalization dataset), RFMS is over 70x faster than MS++. On the WallRobots dataset, where both $d$ and $n$ are large, RFMS show the overall best efficiency performance. The results above demonstrate the good scalability and mode-seeking quality of RFMS in comparison with previous mean-shift approaches.

### 7.2 PIXEL-LEVEL IMAGE SEGMENTATION

Mean-shift is also a popular vision algorithm commonly used for pixel-level segmentation [15; 38]. It is useful in generating initial region proposals or superpixels for deep semantic segmentation networks [55; 6; 45; 59; 77; 80]. Adapting a similar evaluation setup as MS++ [38], we conduct experiments on the Berkeley Segmentation Dataset Benchmark (BSDS500) [56], which contains 500 images with human-labelled segments.

| Method | WallRobot $d = 4, n = 5456, D = 2000$ | | UserKnowledge $d = 5, n = 403, D = 500$ | | WirelessLocalization $d = 7, n = 2000, D = 750$ | |
|--------|------|---------|------|---------|------|---------|
| | NMI | Time(s) | NMI | Time(s) | NMI | Time(s) |
| **MS** | $33.8 \pm 0.0$ | $32.9 \pm 2.8$ | $34.3 \pm 0.0$ | $0.2 \pm 0.0$ | $74.5 \pm 0.0$ | $4.5 \pm 0.6$ |
| **MS++** | $33.5 \pm 0.0$ | $27.7 \pm 1.2$ | $32.6 \pm 0.0$ | $8.6 \pm 0.5$ | $70.8 \pm 0.0$ | $499.8 \pm 129.5$ |
| **RFMS** | $31.4 \pm 0.7$ | $9.5 \pm 0.3$ | $33.6 \pm 1.7$ | $2.8 \pm 0.1$ | $71.1 \pm 2.9$ | $6.9 \pm 1.5$ |

Table 1: Comparison of different mean-shift algorithms on additional higher-dimensional real datasets.

Each image is processed into a dataset containing 154401 three-dimensional points representing pixels in LAB color space. In addition to MS, BMS, and MS++ baselines, we also include QuickShift [69]; another popular segmentation algorithm based on mean-shift that jointly considers spatial and color features. We use the blurring version of RFMS. Due to the inefficiency of the conventional mean-shift algorithm, MS and BMS were run on images $1/36$ of the original size, all other methods were run on full resolution. For RFMS, we set $D = 10$. All methods were run until convergence or a maximum of 100 iterations.

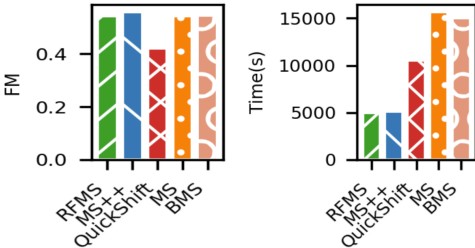

Figure 6: Comparison of different mean-shift image segmentation algorithms in terms of time and Fowlkes-Mallows Score(FM). Example segmented images can be found in Appendix O.

**Results:** We observe that RFMS, MS++, MS, and BMS all perform equally well on segmentation tasks. Despite MS and BMS being run on lower-resolution sampled images, RFMS still achieves 3x speedup when compared with MS and BMS, and 2x speedup when compared to QuickShift. In contrast to the clustering experiments, we also observed that RFMS is slightly faster than MS++. This is due to the fact that the MS++ algorithm does not scale well to higher-dimensional input because the number of neighboring hypercubes increases exponentially with dimensionality. RFMS, however, is not affected by the dimension of the data.

**Additional information**: Full experimental details regarding baseline algorithms, important hyperparameters, and additional results can be found in Appendix N and O. We also provide a comprehensive ablation study regarding the sensitivity of RFMS hyperparameters $(D, T, h, \eta, \mu)$ and the effects of using MC and QMC sampling for RFMS in Appendix M. Furthermore, we also provided useful discussions on the significance of mean-shift algorithms, limitations & future works of RFMS in Appendix C and D.

# 8 CONCLUSION

Mean-shift is the de facto algorithm for mode-seeking - a fundamental procedure in many areas. In this paper, we propose Random Feature mean-shift (RFMS) for mode-seeking over kernel density estimation. Built on top of Random Feature method and zeroth-order optimization, RFMS is an asymptotic complexity improvement over the classical mean-shift algorithm. Theoretically, we show that the modes RFMS produces are close to the actual modes of the kernel density estimation, making RFMS an effective and efficient mode-seeking algorithm. Rather than presenting standard RFF concentration bounds, we develop a complete pipeline tailored to mode seeking. The key significance and novelty here is connecting random-feature approximation to the preservation of modes and mode-seeking dynamics, which, to our knowledge, is not addressed by prior RFF analyses that focus on kernel/value approximation. Empirically, RFMS matches the best clustering NMI within 0.03 while delivering up to 12× speedups. Similarly, on BSDS500, it attains 2–3× speedup compared to the baselines. This advancement broadens the practical applicability of mean-shift algorithms to domains previously limited by high computational demands.

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

The appendix here provides additional details for the ICLR 2026 submission, titled "Random Feature Mean-Sift". The appendix is organized as follows:

- **A - List of Notation**

- **B - RFMS Density Estimation Visualization**

- **C - Discussion**

- **D - Limitation & Future Work**

- **E. Proof of Equation 4**

- **F - Proof of Theorem 1**

- **G - Proof of Theorem 2**

- **H - Proof of Theorem 3**

- **I - Proof of Theorem 4**

- **J - Proof of Theorem 5**

- **K - Proof of Theorem 6**

- **L - Proof of Equation 13**

- **M - Ablation Study**

- **N - Additional Details on Clustering Experiments**

- **O - Additional Details on Image Segmentation Experiments**

- **P - Reproducibility / Code Availability**

- **Q - LLM Usage**

# A    LIST OF NOTATION

We hereby provide a list of notations used in this paper and accompanying proofs:

| Symbol | Meaning |
|---|---|
| $d, D$ | Data dimension, number of random feature |
| $C$ | Universal constant |
| $x$ | data point in $\mathbb{R}^d$ |
| $\mathcal{D}$ | Dataset |
| $n$ | Number of data points in $\mathcal{D}$ |
| $k$ | Positive symmetric kernel function. |
| $h$ | Gaussian kernel bandwidth |
| $k_h$ | Gaussian kernel of bandwidth $h$ |
| $f$ | Data generating density function for $\mathcal{D}$ |
| $\hat{f}_{k_h}$ | Kernel density estimation of $f$ using $k_h$ |
| $c$ | Normalizing constant making sure the integral of $\hat{f}_{k_h}$ is 1 |
| $x_i^{(l)}$ | Point $x_i$ aftter $l$ iterations of mean-shift |
| $\phi_{k_h}$ | Random Feature transformation for kernel $k_h$ |
| $\nabla \hat{f}_{k_h}, \hat{\nabla} \hat{f}_{k_h}$ | Gradient and Estimated gradient of KED |
| $\Phi$ | Summation of random feature transformed points |
| $\hat{g}$ | Random Feature estimation of KDE |
| $\omega$ | Frequencies for constructing Random Feature Mapping |
| $T$ | Total number if iteration |
| $\eta$ | Learning rate for gradient ascent |
| $\mu$ | Smoothing parameter in zeroth-order optimization |
| $j$ | Imaginary unit |
| $\mathcal{X}$ | Compact set over $\mathbb{R}^d$ |
| $\text{Lip}(\cdot)$ | Lipschitz constant of a function |
| $\lambda_{min}(\cdot)$ | Smallest eigenvalue of a square matrix |
| $B(x, r)$ | Closed ball centered at $x$ with radius $r$ |
| $\theta$ | Łojasiewicz exponent |
| $\|\cdot\|$ | $L^2$ norm |
| $\|\cdot\|_{\psi_1}$ | Sub-exponential Orlicz norm |
| $\|\cdot\|_F$ | Frobenius norm |
| $\|\cdot\|_{op}$ | Operator norm |

Table 2: List of notations.

# B    RFMS DENSITY ESTIMATION VISUALIZATION

This section provides a qualitative sanity check of the random-feature density approximation used by RFMS. While Theorems 3–5 establish that the approximation preserves mode locations up to a small perturbation (and that any spurious modes must be confined to low-density regions), visualizing the estimated density surfaces offers an intuitive confirmation of these claims.

**Setup:** We generate synthetic 1D and 2D mixtures of Gaussians with multiple separated (and mildly overlapping) components, so that the ground-truth KDE exhibits several distinct modes. For each dataset, we compute (i) the exact Gaussian KDE and (ii) the RFF density approximation obtained from the same kernel bandwidth but replacing the kernel evaluation with a finite-dimensional random-feature map of dimension $D$. We visualize the resulting density functions on a uniform grid (line plot in 1D; 3D surface and heatmap in 2D). The results are shown in Figure 7.

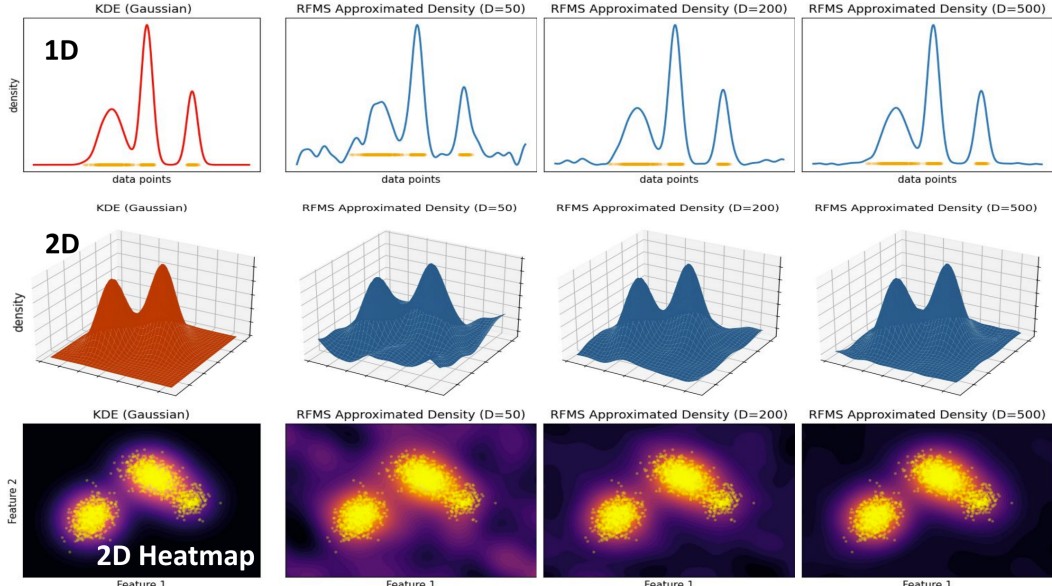

Figure 7: Density visualization on synthetic 1D and 2D points using Gaussian KDE and random feature density estimation. (Top) Actual and RFF approximated density function using 800 1D points with 3 clusters and different variances. (Middle) Actual and RFF approximated density function using 2000 2D points with 2 clusters and different variances. (Bottom) Heatmap of 2D examples.

**Observations:** Across both 1D and 2D examples, the RFMS desnity approximation accurately reproduces the dominant basins of attraction and preserves the number and locations of the high-density modes. Small local discrepancies mostly visible in the tails where the true density is near zero. In these low-density regions, the RFMS surface may exhibit mild ripples; consistent with our theory, such oscillations can introduce visually small, isolated extrema that do not correspond to meaningful modes and do not affect mode-seeking trajectories initialized in moderate-to-high density regions. As $D$ increases, the RFMS surface becomes progressively smoother and converges visually to the KDE surface: peak locations stabilize, and tail oscillations diminish. This qualitative trend aligns with the approximation guarantees, where the gradient/Hessian error decreases with larger $D$, implying improved stability of critical points and their local geometry. Overall, these visualizations support the theoretical guarantees that RFMS preserves the relevant mode structure of the KDE and that any approximation-induced artifacts are limited to low-density regions where they have minimal impact on the practical mode-seeking behavior.

## C DISCUSSION

### C.1 COMPARISON WITH OTHER FAST KDE APPROACHES

A rich body of literature accelerates KDE through techniques such as coresets, locality sensitive hashing (LSH), and specialized data structures (e.g., space partitioning) [63; 10; 43; 27; 73; 39; 48]. Our focus here is different in that we aim for an end-to-end mode-seeking algorithm whose per-iteration cost is linear in $n$, without discretizing the domain or relying on search structures that can weaken in higher dimensions (e.g., tree, grid–style space partitions, or fast Gauss transform-based approaches [27; 73; 39; 48] whose cost grows rapidly with ambient data dimension). More importantly, those prior methods primarily target fast evaluation of KDE at a query point, whereas our analysis is designed to support mode-seeking correctness: we bound how the modes of the approximate KDE move relative to true KDE modes (Theorems 3–5) and prove convergence of a stochastic/ascent-style iteration to a neighborhood of those modes (Theorem 6). Propagating RFF-based KDE approximation error through to mode stability and tracking guarantees is a central

contribution of our work. We also note a compatibility issue with blurring mean shift: when points move each iteration, data-structure or coreset-based accelerations may require rebuilding or substantial updates, potentially eroding their efficiency. In contrast, RFMS only updates and sums the evolving feature encodings, which remains inexpensive and highly parallelizable. Overall, we view RFMS as a complementary point in the broader design space of fast KDE approximations, alongside coresets, LSH, and partitioning methods. Distinguished by RFMS's simplicity in computation and a well-established RFF theoretical foundation.

## C.2 COMPATIBLE KERNEL FUNCTIONS

We note that the class of shift-invariant kernels is extensive — including Gaussian, Laplacian, Cauchy, Matérn, and other widely used non-negative similarity measurements. Furthermore, RFMS is not restricted to Fourier-based kernels. It can accommodate other random feature constructions, such as Polynomial kernels via random Maclaurin expansions or dot-product kernels. In this sense, RFMS is a general mode-seeking framework over random feature approximated kernel densities, and is not inherently limited to any particular kernels.

## C.3 BOUNDS TIGHTNESS

Theorem 1 attains the canonical Monte Carlo RFF rate where the KDE approximation error decays as $\mathcal{O}(1/\sqrt{D})$ [65] in the number of features $D$. This dependence is standard (and essentially optimal) for vanilla i.i.d. feature sampling. Theorems 3-4 extend the same $\mathcal{O}(1/\sqrt{D})$ dependence to the gradient and Hessian. The extra factors are the usual cost of controlling the derivatives uniformly, not a deterioration in the RFF sampling rate. We note that the constants in the bounds can be conservative due to the use of Hoeffding-type inequality arguments, but the rates in $D$ are the key notion of tightness here and are sufficient for our mode-tracking guarantees. Practically, this yields a clean accuracy-efficiency knob; the RFF-KDE is unbiased w.r.t. feature randomness, so increasing $D$ reduces variance and improves approximation, while runtime scales roughly linearly in $D$ and $T$. Thus, the approximation can be made arbitrarily tight by choosing $D$ large enough.

## C.4 RELATION TO HDC/VSA

Hyperdimensional computing (HDC)—also known as Vector Symbolic Architecture (VSA)— is a class of computational models that represent and manipulate structured information using high-dimensional vectors. The characteristic of VSA/HDC is that it first encodes data as high-dimensional vectors and operates on encoded data using a set of simple algebraic operations that are efficient and highly parallelizable [41; 64]. HDC is connected to theoretical neuroscience as its mathematical framework closely resembles models of neural coding in the brain [22]. As a result, various machine learning algorithms based on HDC/VSA have been proposed, such as classification [23], clustering [24], and regression [31]. However, there is a lack of density-based analysis methods in existing HDC/VSA literature. The notion of high-dimensional, distributed, and compositional representation of HDC/VSA aligns closely with RFMS. In that sense, RFMS can be viewed functionally as an HDC/VSA algorithm, therefore filling in the gap between HDC/VSA and density-based analysis. Furthermore, HDC/VSA has also been extensively studied, especially within the hardware community. Various types of accelerators [79; 67] have been proposed for HDC/VSA workloads. This explicit connection between RFMS and HDC/VSA, and their computational similarity, can potentially lead to the use of existing HDC/VSA accelerators for RFMS, providing practical benefits.

## C.5 SIGNIFICANCE OF MEAN-SHIFT ALGORITHMS

Mean-Shift, as a geometry-respecting procedure, is broadly useful across machine learning and data analysis, some notable examples including vision, anomaly detection, self-supervised learning and more [38; 44; 75]. Despite known limitations such as bandwidth sensitivity and the applicability in high-dimensional data, mean-shift remains meaningful because it is often used as an algorithmic primitive and is still being actively used by recent research [2; 53; 44]. Furthermore, the regimes where mean-shift performs well are well understood [25; 15], and our approach provides a faithful

approximation with asymptotic complexity improvements, making it desirable for many existing and emerging applications.

# D   LIMITATION & FUTURE WORK

## D.1   MORSE FUNCTION ASSUMPTION

In Theorem 5, our results regarding the stability of critical points have the assumption that the KDE is a Morse function, meaning a smooth function with non-degenerate critical points. This is, in fact, a standard assumption in density-based mode analysis [1; 11]. Moreover, the general body of literature regarding the mean-shift algorithm assumes that gradient ascent on a KDE surface is well-behaved. This implicitly assumes the density function has isolated, non-degenerate modes, aligning with the Morse function.

## D.2   SEMANTIC SEGMENTATION

In the context of image processing, mean-shift is a non-parametric unsupervised algorithm based on low-level features such as color and/or spatial proximity. It operates on the pixel level and groups pixels based on local density in a feature space, not on high-level semantic categories. Consequently, mean-shift can segment coherent regions but cannot segment regions based on semantic information. Despite its limitations, mean-shift can still be helpful in roles like generating superpixels or region proposals, which can be a key step in semantic segmentation with deep neural networks [55; 6; 45; 59; 77].

## D.3   MODE-SEEKING FOR HIGH-DIMENSIONAL DATA

In the high-dimensional regime, both RFMS and classical mean shift (or any KDE-based method) are fundamentally limited by the curse of dimensionality. Our goal here is not to fix this statistical issue, but to provide a computationally scalable approximation to classical mean-shift in regimes where it is still used, which leads to substantial runtime gains in large $n$ settings for the low-to-moderate dimensional data. Despite this limitation, it remains useful and a powerful primitive in many applications [38; 44; 75; 2; 53; 44; 14]. We believe RFMS can be incorporated with other methods (e.g., dimensionality reduction methods such as in [54; 49]) and be explored in higher-dimensional regimes. Our present goal, however, is to lay the theoretical foundations and analyze frameworks for this sampling-based mean-shift estimator to support any future extensions.

## D.4   FUTURE WORK

The primary aim of this paper is to establish the algorithmic and theoretical foundation of RFMS. We view this work as a principled first step toward scalable, kernel-based mode-seeking for domains such as tracking and point cloud, as these applications have been addressed by inefficient forms of mean-shift [37]. RFMS is also designed to be modular and extensible: different random-feature maps and zeroth-order gradient strategies can be used depending on the application. We believe RFMS can be incorporated into other learning pipelines (e.g., deep neural networks) [2; 53; 44]. It is also well known that the mean-shift algorithms work best in a low-to-moderate dimension regime, as kernel density estimation suffers from the curse of dimensionality [25]. As a potential future direction, we would also like to extend RFMS into high-dimensional regimes [10; 9]. In any case, this paper can serve as a theoretical foundation for this sampling-based mean-shift estimator in support of any future extensions.

# E   PROOF OF EQUATION 4

*Proof.*   Want to show $\hat{g}(x)$ is an unbiased estimation of the kernel density estimation $\hat{f}(x)$ with respect to the randomness in $\omega \sim \mathcal{N}\left(0, \frac{1}{h^2}\mathbf{I}_d\right)$. Recall the definition of $\phi_{k_h}$:

$$\phi_{k_h} = \frac{1}{\sqrt{D}} \left[ e^{j(\omega_1^T x)} e^{j(\omega_2^T x)} e^{j(\omega_3^T x)} \ldots e^{j(\omega_D^T x)} \right] \tag{15}$$

We start by showing that the complex inner products between encodings produced by $\phi_{k_h}$ are an unbiased estimation of the kernel function $k_h$. For any $x, x' \in \mathbb{R}^d$, the results follow directly after linearity of expectation and Bochner's theorem:

$$
\begin{aligned}
&\mathbb{E}_{\omega \sim \mathcal{N}\left(0, \frac{1}{h^2}\mathbf{I}_d\right)} \left[ \langle \phi_{k_h}(x), \phi_{k_h}(x') \rangle \right] \\
&= \mathbb{E}_{\omega \sim \mathcal{N}\left(0, \frac{1}{h^2}\mathbf{I}_d\right)} \left[ \frac{1}{D} \sum_{k=1}^{D} \overline{e^{j(\omega_k^T x)}} e^{j(\omega_k^T x')} \right] \\
&= \mathbb{E}_{\omega \sim \mathcal{N}\left(0, \frac{1}{h^2}\mathbf{I}_d\right)} \left[ \overline{e^{j(\omega^T x)}} e^{j(\omega^T x')} \right] \\
&= \mathbb{E}_{\omega \sim \mathcal{N}\left(0, \frac{1}{h^2}\mathbf{I}_d\right)} \left[ e^{j\omega^T (x'-x)} \right] \\
&= \int_{\mathbb{R}^d} p(\omega) e^{j\omega \cdot (x'-x)} d\omega \\
&= k_h(x', x) = k_h(x, x')
\end{aligned}
\tag{16}
$$

Recall that $\Phi = \frac{1}{nc} \sum_{i=1}^{n} \phi_{k_h}(x_i) \forall x_i \in \mathcal{D}$ and $\hat{g}(x) = \langle \phi_{k_h}(x), \Phi \rangle$, so:

$$
\begin{aligned}
&\mathbb{E}_{\omega \sim \mathcal{N}\left(0, \frac{1}{h^2}\mathbf{I}_d\right)} \left[ \langle \phi_{k_h}(x), \Phi \rangle \right] \\
&= \mathbb{E}_{\omega \sim \mathcal{N}\left(0, \frac{1}{h^2}\mathbf{I}_d\right)} \left[ \overline{\phi_{k_h}(x)} \cdot \left( \frac{1}{nc} \sum_{i=1}^{n} \phi_{k_h}(x_i) \right) \right] \\
&= \mathbb{E}_{\omega \sim \mathcal{N}\left(0, \frac{1}{h^2}\mathbf{I}_d\right)} \left[ \frac{1}{nc} \sum_{i=1}^{n} \left( \overline{\phi_{k_h}(x)} \cdot \phi_{k_h}(x_i) \right) \right] \\
&= \frac{1}{nc} \sum_{i=1}^{n} k_{k_h}(x, x_i) = \hat{f}_{k_h}(x)
\end{aligned}
\tag{17}
$$

$\blacksquare$

## F   PROOF OF THEOREM 1

*Proof.* First expand $\left| \hat{g}(x) - \hat{f}_{k_h}(x) \right|$:

$$
\begin{aligned}
\left| \hat{g}(x) - \hat{f}_{k_h}(x) \right| &= \left| \langle \phi_{k_h}(x), \Phi \rangle - \frac{1}{nc} \sum_{i=1}^{n} k_h(x, x_i) \right| \\
&= \left| \left\langle \phi_{k_h}(x), \frac{1}{nc} \sum_{i=1}^{n} \phi_{k_h}(x_i) \right\rangle - \frac{1}{nc} \sum_{i=1}^{n} k_h(x, x_i) \right| \\
&= \left| \frac{1}{nc} \sum_{i=1}^{n} \langle \phi_{k_h}(x), \phi_{k_h}(x_i) \rangle - \frac{1}{nc} \sum_{i=1}^{n} k_h(x, x_i) \right| \\
&= \frac{1}{nc} \left| \sum_{i=1}^{n} \langle \phi_{k_h}(x), \phi_{k_h}(x_i) \rangle - \sum_{i=1}^{n} k_h(x, x_i) \right| \\
&= \frac{1}{nc} \left| \sum_{i=1}^{n} \frac{1}{D} \sum_{k=1}^{D} e^{j\omega_k^T (x_i - x)} - \sum_{i=1}^{n} k_h(x, x_i) \right|
\end{aligned}
\tag{18}
$$

We define an estimator:

$$Z_k = \sum_{i=1}^{n} e^{j\omega_k^T(x_i - x)} \tag{19}$$

Since Random Feature is unbiased for approximating individual kernel values, we know that:

$$\mathbb{E}_{\omega \sim \mathcal{N}\left(0, \frac{1}{h^2}\mathbf{I}_d\right)}[Z_k] = \sum_{i=1}^{n} k_h(x, x_i) \tag{20}$$

Since the sampling of random features $\omega$ in the construction of $\phi_{k_h}$ is uniform i.i.d., the approximation is an average of i.i.d. complex-valued random variables $Z_k$, with expectation equal to the target kernel sum. Split $Z_k$ into real and imaginary parts using sine and cosine:

$$Z_k = A_k + jB_k \quad \text{where:} \quad A_k = \sum_{i=1}^{n} \cos\left(\omega_k^T(x_i - x)\right), B_k = \sum_{i=1}^{n} \sin\left(\omega_k^T(x_i - x)\right) \tag{21}$$

Since sine and cosine functions are bounded between $[-1, 1]$, it is clear that $A_k, B_k \in [-n, n]$, which is bounded. Use Hoeffding's inequality for both $A_k$ and $B_k$. Chase any $\delta \in (0, 1)$ with probability if at least $1 - \delta/2$:

$$\left| \frac{1}{D}\sum_{k=1}^{D} A_k - \mathbb{E}_{\omega \sim \mathcal{N}\left(0, \frac{1}{h^2}\mathbf{I}_d\right)}[A_k] \right| \leq 2n\sqrt{\frac{1}{2D}\ln\frac{4}{\delta}}$$

$$\left| \frac{1}{D}\sum_{k=1}^{D} B_k - \mathbb{E}_{\omega \sim \mathcal{N}\left(0, \frac{1}{h^2}\mathbf{I}_d\right)}[B_k] \right| \leq 2n\sqrt{\frac{1}{2D}\ln\frac{4}{\delta}} \tag{22}$$

Since $k_h$ is real valued, meaning $\mathbb{E}_{\omega \sim \mathcal{N}\left(0, \frac{1}{h^2}\mathbf{I}_d\right)}[B_k] = 0$, so $\mathbb{E}[A_k] = \mathbb{E}_{\omega \sim \mathcal{N}\left(0, \frac{1}{h^2}\mathbf{I}_d\right)}[Z_k] = \sum_{i=1}^{n} k_h(x, x_i)$. Recall that for complex number $Z_k$, $|Z_k| = \sqrt{A_K^2 + B_K^2} \leq |A_k| + |B_k|$, and the probability of both the imaginary and real inequality holds is $(1 - \delta/2)^2 > 1 - \delta$, so:

$$\left| \frac{1}{D}\sum_{k=1}^{D} Z_k - \sum_{i=1}^{n} k_h(x, x') \right| \leq \left| \frac{1}{D}\sum_{k=1}^{D} A_k - \sum_{i=1}^{n} k_h(x, x_i) \right| + \left| \frac{1}{D}\sum_{k=1}^{D} B_k \right|$$

$$\leq 4n\sqrt{\frac{1}{2D}\ln\frac{4}{\delta}} \tag{23}$$

Finally:

$$\left| \hat{g}(x) - \hat{f}_{k_h}(x) \right| = \frac{1}{nc}\left| \frac{1}{D}\sum_{k=1}^{D} Z_k - \sum_{i=1}^{n} k_h(x, x') \right|$$

$$\leq \frac{4}{c}\sqrt{\frac{1}{2D}\ln\frac{4}{\delta}} \quad \text{With probability at least } 1 - \delta \tag{24}$$

$\blacksquare$

So, for any $\delta \in (0, 1)$, with probability at least $1 - \delta$, the above error bound holds. Additionally, suppose KED are estimated via $\text{Re}(\hat{g}(x))$. In that case, there will be no additional variance from the imaginary part, further reducing the bound to $\frac{2}{c}\sqrt{\frac{1}{2D}\ln\frac{4}{\delta}}$.

## G   PROOF OF THEOREM 2

*Proof.* Let $\mathcal{X}$ be a compact set over $\mathbb{R}^d$ such that $\mathcal{D} \subset \mathcal{X}$. The goal is to extend Theorem 1 to a uniform convergence statement over the entire $\mathcal{X}$.

$$
\begin{aligned}
\sup_{x \subset \mathcal{X}} \left| \left( \operatorname{Re}\left(\hat{g}(x)\right) - \hat{f}_{k_h}(x) \right) \right| &= \sup_{x \subset \mathcal{X}} \left| \frac{1}{nc} \left\| \sum_{i=1}^{n} \operatorname{Re}\left(\langle \phi_{k_h}(x), \phi_{k_h}(x_i) \rangle\right) - \sum_{i=1}^{n} k_h\left(x, x_i\right) \right\| \right. \\
&\leq \frac{1}{c} \sup_{x, y \subset \mathcal{X}} \left\| \operatorname{Re}\left(\langle \phi_{k_h}(x), \phi_{k_h}(y) \rangle\right) - k_h\left(x, y\right) \right\|
\end{aligned}
\tag{25}
$$

Ignoring the constant $\frac{1}{c}$ for now, the remaining is the uniform convergence of random Fourier features, which has been studied by Rahimi and Recht [65], for an error tolerance epsilon $\epsilon$:

$$
\begin{aligned}
\Pr &\left( \sup_{x, y \subset \mathcal{X}} \left\| \operatorname{Re}\left(\langle \phi_{k_h}(x), \phi_{k_h}(y) \rangle\right) - k_h\left(x, y\right) \right\| \geq \epsilon \right) \\
&\leq 2^8 \left( \frac{\sqrt{\mathbb{E}_{\omega \sim \mathcal{N}\left(0, \frac{1}{h^2}\mathbf{I}_d\right)}\left[\langle \omega, \omega \rangle\right]}\operatorname{diam}(\mathcal{X})}{\epsilon} \right)^2 \exp\left( -\frac{D\epsilon^2}{4(d+2)} \right)
\end{aligned}
\tag{26}
$$

Next, solve $\sqrt{\mathbb{E}_{\omega \sim \mathcal{N}\left(0, \frac{1}{h^2}\mathbf{I}_d\right)}\left[\langle \omega, \omega \rangle\right]}$:

$$
\sqrt{\mathbb{E}_{\omega \sim \mathcal{N}\left(0, \frac{1}{h^2}\mathbf{I}_d\right)}\left[\langle \omega, \omega \rangle\right]} = \sqrt{\frac{d}{h^2}} = \frac{\sqrt{d}}{h}
\tag{27}
$$

Putting everything back together:

$$
\Pr\left( \sup_{x \subset \mathcal{X}} \left| \operatorname{Re}\left(\hat{g}(x)\right) - \hat{f}_{k_h}(x) \right| \geq \epsilon \right) \leq 2^8 \left( \frac{c\sqrt{d}\operatorname{diam}(\mathcal{X})}{h\epsilon} \right)^2 \exp\left( -\frac{D\epsilon^2}{c^2 4(d+2)} \right)
\tag{28}
$$

∎

## H   PROOF OF THEOREM 3

*Proof.* Here, we would like to show a bound between the gradient of the Random Feature approximated KDE and the gradient of the actual KDE. Specifically, since we are using real-valued version of $\hat{g}(x)$, we what to show a bound on $\left\| \nabla \operatorname{Re}\left(\hat{g}(x)\right) - \nabla \hat{f}_{k_h}(x) \right\|$. We first expand:

$$
\begin{aligned}
\left\| \nabla \operatorname{Re}\left(\hat{g}(x)\right) - \nabla \hat{f}_{k_h}(x) \right\| &= \frac{1}{nc} \left\| \sum_{i=1}^{n} \frac{1}{D} \sum_{k=1}^{D} \nabla \operatorname{Re}\left(e^{j\omega_k^T(x_i - x)}\right) - \sum_{i=1}^{n} \nabla k_h\left(x, x_i\right) \right\| \\
&= \frac{1}{nc} \left\| \frac{1}{D} \sum_{k=1}^{D} \sum_{i=1}^{n} \nabla \cos(\omega_k^T(x_i - x)) - \sum_{i=1}^{n} \nabla k_h\left(x, x_i\right) \right\|
\end{aligned}
\tag{29}
$$

We define an estimator:

$$
Z_k = \sum_{i=1}^{n} \nabla \cos(\omega_k^T(x_i - x))
\tag{30}
$$

Since Differentiation commutes with expectation:

$$\mathbb{E}_{\omega\sim\mathcal{N}\left(0,\frac{1}{h^2}\mathbf{I}_d\right)}[Z_k] = \nabla\mathbb{E}_{\omega\sim\mathcal{N}\left(0,\frac{1}{h^2}\mathbf{I}_d\right)}\left[\sum_{i=1}^{n}\cos(\omega_k^T(x_i-x))\right] = \sum_{i=1}^{n}\nabla k_h(x,x_i) \quad (31)$$

To bound the deviation of $Z_k$ from its expectation, we apply Bernstein's inequality:

$$Z_k = \sum_{i=1}^{n}\sin\left(\omega_k^T(x_i-x)\right)\omega_k \quad (32)$$

However, since $\omega\sim\mathcal{N}\left(0,\frac{1}{h^2}\mathbf{I}_d\right)$ is unbounded, random variable $Z_k$ is also unbounded, so the standard Bernstein inequality does not apply. However, notice that the $\omega$ is sub-exponential, we apply the Bernstein inequality for a tail-heavy random variable.

Consider the version of Bernstein inequality presented by Lanthaler et al. [46] which states: Let $Z$ be a sub-exponential random variable in a separable Hilbert space, choose any $\delta\in(0,1)$, with probability at least $1-\delta$, the following holds:

$$\left\|\frac{1}{D}\sum_{k=1}^{D}Z_k - \mathbb{E}_{\omega\sim\mathcal{N}\left(0,\frac{1}{h^2}\mathbf{I}_d\right)}[Z_k]\right\| \leq \frac{2b\ln(2/\delta)}{D} + \sqrt{\frac{2\sigma^2\ln(2/\delta)}{D}} \quad (33)$$

Where:

$$\sigma^2 = 4e\sqrt{\mathbb{E}_{\omega\sim\mathcal{N}\left(0,\frac{1}{h^2}\mathbf{I}_d\right)}\left[\|Z_k-\mathbb{E}_{\omega\sim\mathcal{N}\left(0,\frac{1}{h^2}\mathbf{I}_d\right)}[Z_k]\|^2\right]}\|Z_k\|_{\psi_1} \quad \text{and} \quad b = 4e\|Z_k\|_{\psi_1} \quad (34)$$

$\|\cdot\|_{\psi_1}$ denotes sub-exponential Orlicz norm. Since $\mathbb{R}^d$ is a separable Hilbert space, it is directly applicable here. First bounding:

$$\begin{aligned}
&\mathbb{E}_{\omega\sim\mathcal{N}\left(0,\frac{1}{h^2}\mathbf{I}_d\right)}\left[\left\|Z_k-\mathbb{E}_{\omega\sim\mathcal{N}\left(0,\frac{1}{h^2}\mathbf{I}_d\right)}[Z_k]\right\|^2\right] \\
&= \mathbb{E}_{\omega\sim\mathcal{N}\left(0,\frac{1}{h^2}\mathbf{I}_d\right)}\left[\|Z_k\|^2\right] - \|\mathbb{E}_{\omega\sim\mathcal{N}\left(0,\frac{1}{h^2}\mathbf{I}_d\right)}[Z_k]\|^2 \\
&\leq \mathbb{E}_{\omega\sim\mathcal{N}\left(0,\frac{1}{h^2}\mathbf{I}_d\right)}\left[\|Z_k\|^2\right]
\end{aligned} \quad (35)$$

Since the sine function is bounded:

$$\|Z_k\| \leq n\|\omega_k\| \quad (36)$$

Recall that $\omega_k$ are drawn from $\mathcal{N}\left(0,\frac{1}{h^2}\mathbf{I}_d\right)$, so:

$$\mathbb{E}_{\omega\sim\mathcal{N}\left(0,\frac{1}{h^2}\mathbf{I}_d\right)}\left[\|Z_k\|^2\right] \leq n^2\mathbb{E}_{\omega\sim\mathcal{N}\left(0,\frac{1}{h^2}\mathbf{I}_d\right)}\left[\|\omega_k\|^2\right] \leq \frac{n^2 d}{h^2} \quad (37)$$

Now bounding $\|Z_k\|_{\psi_1}$:

$$\|Z_k\|_{\psi_1} = \|\|Z_k\|\|_{\psi_1} \leq \|n\|\omega_k\|\|_{\psi_1} = n\|\|\omega_k\|\|_{\psi_1} = n\|\omega_k\|_{\psi_1} = \frac{n}{h}\|z\|_{\psi_1} \quad (38)$$

Where $z\sim\mathcal{N}(0,I)$. We know that [70]:

$$\|z\|_{\psi_1} \leq C\sqrt{d} \quad (39)$$

Where $C$ is a universal constant, so:

$$\|Z_k\|_{\psi_1} \leq \frac{Cn\sqrt{d}}{h} \tag{40}$$

Which means:

$$\sigma^2 \leq \frac{4eCn^2 d}{h^2} \quad \text{and} \quad b \leq \frac{4eCn\sqrt{d}}{h} \tag{41}$$

We can bound the deviation of $Z_k$ from its expectation:

$$\left\| \frac{1}{D} \sum_{k=1}^{D} Z_k - \mathbb{E}_{\omega \sim \mathcal{N}\left(0, \frac{1}{h^2}\mathbf{I}_d\right)} [Z_k] \right\| \leq \frac{8Cen\sqrt{d}\ln(2/\delta)}{Dh} + \sqrt{\frac{8Cen^2 d \ln(2/\delta)}{Dh^2}} \tag{42}$$

The above holds with probability at least $1 - \delta$. And finally, also with probability at least $1 - \delta$:

$$\left\| \nabla \operatorname{Re}(\hat{g}(x)) - \nabla \hat{f}_{k_h}(x) \right\| \leq \frac{1}{nc} \left( \frac{8Cen\sqrt{d}\ln(2/\delta)}{Dh} + \sqrt{\frac{8Cen^2 d \ln(2/\delta)}{Dh^2}} \right) \tag{43}$$

∎

## I  PROOF OF THEOREM 4

*Proof.* We also want to bound the Frobenius norm of the difference between the Hessian:

$$\left\| \nabla^2 \operatorname{Re}(\hat{g}(x)) - \nabla^2 \hat{f}_{k_h}(x) \right\|_F = \frac{1}{nc} \left\| \frac{1}{D} \sum_{k=1}^{D} \sum_{i=1}^{n} \nabla^2 \cos(\omega_k^T (x_i - x)) - \sum_{i=1}^{n} \nabla^2 k_h(x, x_i) \right\|_F \tag{44}$$

Define an estimator:

$$Z_k = \sum_{i=1}^{n} \nabla^2 \cos(\omega_k^T (x_i - x)) \tag{45}$$

Again, because differentiation commutes with expectation, know that:

$$\mathbb{E}_{\omega \sim \mathcal{N}\left(0, \frac{1}{h^2}\mathbf{I}_d\right)} [Z_k] = \nabla^2 \mathbb{E}_{\omega \sim \mathcal{N}\left(0, \frac{1}{h^2}\mathbf{I}_d\right)} \left[ \sum_{i=1}^{n} \cos(\omega_k^T (x_i - x)) \right] = \sum_{i=1}^{n} \nabla^2 k_h(x, x_i)$$

$$\tag{46}$$

Use the proving technique as seen in the proof of Theorem 3. We start by bounding $\sigma^2$ and $b$:

$$Z_k = \sum_{i=1}^{n} - \cos\left(\omega_k^T (x_i - x)\right) \omega_k \omega_k^T \tag{47}$$

Then:

$$\mathbb{E}_{\omega \sim \mathcal{N}\left(0, \frac{1}{h^2}\mathbf{I}_d\right)}\left[\left\|Z_k - \mathbb{E}_{\omega \sim \mathcal{N}\left(0, \frac{1}{h^2}\mathbf{I}_d\right)}\left[Z_k\right]\right\|_F^2\right]$$
$$= \mathbb{E}_{\omega \sim \mathcal{N}\left(0, \frac{1}{h^2}\mathbf{I}_d\right)}\left[\|Z_k\|_F^2\right] - \left\|\mathbb{E}_{\omega \sim \mathcal{N}\left(0, \frac{1}{h^2}\mathbf{I}_d\right)}[Z_k]\right\|_F^2 \tag{48}$$
$$\leq \mathbb{E}_{\omega \sim \mathcal{N}\left(0, \frac{1}{h^2}\mathbf{I}_d\right)}\left[\|Z_k\|_F^2\right]$$

Since the cosine function is bounded:

$$\|Z_k\|_F \leq n\|\omega_k \omega_k^T\|_F \tag{49}$$

Recall that $\omega_k$ are drawn from $\mathcal{N}\left(0, \frac{1}{h^2}\mathbf{I}_d\right)$, so:

$$\mathbb{E}_{\omega \sim \mathcal{N}\left(0, \frac{1}{h^2}\mathbf{I}_d\right)}\left[\|Z_k\|_F^2\right] \leq n^2 \mathbb{E}_{\omega \sim \mathcal{N}\left(0, \frac{1}{h^2}\mathbf{I}_d\right)}\left[\|\omega_k \omega_k^T\|_F^2\right] \leq \frac{n^2 d(d+2)}{h^4} \tag{50}$$

Now bounding $\|Z_k\|_{\psi_1}$:

$$\|Z_k\|_{\psi_1} = \|\|Z_k\|_F\|_{\psi_1} \leq \|n\|\omega_k \omega_k^T\|_F\|_{\psi_1} = n\|\|\omega_k \omega_k^T\|_F\|_{\psi_1} = n\|\omega_k \omega_k^T\|_{\psi_1} \leq \frac{Cnd}{h^2} \tag{51}$$

Which means:

$$\sigma^2 \leq \frac{4eCn^2 d\sqrt{d(d+2)}}{h^4} \quad \text{and} \quad b \leq \frac{4eCnd}{h^2} \tag{52}$$

Bound the deviation of $Z_k$ from its expectation:

$$\left\|\frac{1}{D}\sum_{k=1}^{D} Z_k - \mathbb{E}_{\omega \sim \mathcal{N}\left(0, \frac{1}{h^2}\mathbf{I}_d\right)}[Z_k]\right\|_F \leq \frac{8eCnd\ln(2/\delta)}{Dh^2} + \sqrt{\frac{8eCn^2 d\sqrt{d(d+2)}\ln(2/\delta)}{Dh^4}} \tag{53}$$

The above holds with probability at least $1 - \delta$. And finally, also with probability at least $1 - \delta$:

$$\left\|\nabla^2 \operatorname{Re}(\hat{g}(x)) - \nabla^2 \hat{f}_{k_h}(x)\right\|_F \leq \frac{1}{nc}\left(\frac{8eCnd\ln(2/\delta)}{Dh^2} + \sqrt{\frac{8eCn^2 d\sqrt{d(d+2)}\ln(2/\delta)}{Dh^4}}\right) \tag{54}$$

∎

## J    PROOF OF THEOREM 5

*Proof.* The aim here is to show the critical points between $\operatorname{Re}(\hat{g})$ and $\hat{f}_{k_h}$ are close under a mild assumption. Assume $\hat{f}_{k_h}$ is a Morse function, i.e., a smooth function with non-degenerate critical points [29], let $x^*$ be any critical point of $\hat{f}_{k_h}$, we have:

$$\nabla f_{k_h}(x^*) = 0 \quad \text{and} \quad \nabla^2 f_{k_h}(x^*) \text{ is invertible with all positive eigenvalues} \tag{55}$$

In Theorem 3 and 4, we have shown that $\operatorname{Re}(\hat{g})$ and $\hat{f}_{k_h}$, are point-wise close in terms of their gradient and hessian for all $x \in \mathbb{R}^d$, suppose:

$$\|\nabla \operatorname{Re}(\hat{g}(x)) - \nabla f_{k_h}(x)\| \leq \epsilon_1 \quad \text{With probability at least } 1 - \delta/2$$
$$\left\|\nabla^2 \operatorname{Re}(\hat{g}(x)) - \nabla^2 f_{k_h}(x)\right\|_F \leq \epsilon_2 \quad \text{With probability at least } 1 - \delta/2 \tag{56}$$

Where $\epsilon_1, \epsilon_2$ can be made arbitrarily small by increasing $D$ (see Theorem 3 and Theorem 4).

We're looking for $\hat{x}^*$ near $x^*$ such that:

$$\nabla \operatorname{Re}(\hat{g}(\hat{x}^*)) = 0 \tag{57}$$

$\hat{x}^*$ can be found via Newton iteration:

$$T(x) = x - \left[\nabla^2 \operatorname{Re}(\hat{g}(x))\right]^{-1} \nabla \operatorname{Re}(\hat{g}(x)) \tag{58}$$

Start at $x^*$, we can show the distance between $x^*$ and $\hat{x}^*$ using Newton–Kantorovich theorem [42], which gives the optimality and convergence result of Newton's Method.

We first bound the gradient residual at $x^*$:

$$\|\nabla \operatorname{Re}(\hat{g}(x^*))\| = \left\|\nabla \operatorname{Re}(\hat{g}(x^*)) - \nabla \hat{f}_{k_h}(x^*)\right\| \leq \epsilon_1 \tag{59}$$

Then, show the invertibility of $\nabla^2 \operatorname{Re}(\hat{g}(x^*))$ by showing all its eigenvalue are positive. Let $\lambda_{min}(\cdot)$ denote the smallest eigenvalue of a matrix, by Weyl's inequality:

$$\begin{aligned}
&\left|\lambda_{min}\left(\nabla^2 \operatorname{Re}(\hat{g}(x^*))\right) - \lambda_{min}\left(\nabla^2 \hat{f}_{k_h}(x^*)\right)\right| \\
&\leq \left\|\nabla^2 \operatorname{Re}(\hat{g}(x^*)) - \nabla^2 f_{k_h}(x^*)\right\|_{op} \\
&\leq \left\|\nabla^2 \operatorname{Re}(\hat{g}(x^*)) - \nabla^2 f_{k_h}(x^*)\right\|_F \\
&\leq \epsilon_2
\end{aligned} \tag{60}$$

Which implies:

$$\lambda_{min}\left(\nabla^2 \operatorname{Re}(\hat{g}(x^*))\right) \geq \lambda_{min}\left(\nabla^2 \hat{f}_{k_h}(x^*)\right) - \epsilon_2 \tag{61}$$

This means, in order for $\nabla^2 \operatorname{Re}(\hat{g}(x^*))$ to be invertible, simply chose $\epsilon_2 \leq \lambda_{min}\left(\nabla^2 \hat{f}_{k_h}(x^*)\right)$.
Further more we can bound the operator norm of $\left[\nabla^2 \operatorname{Re}(\hat{g}(x^*))\right]^{-1}$:

$$\left\|\left[\nabla^2 \operatorname{Re}(\hat{g}(x^*))\right]^{-1}\right\|_{op} \leq \frac{1}{\lambda_{min}\left(\nabla^2 \operatorname{Re}(\hat{g}(x^*))\right)} \leq \frac{1}{\lambda_{min}\left(\nabla^2 \hat{f}_{k_h}(x^*)\right) - \epsilon_2} \tag{62}$$

Since we successfully bounded the operator norm, we also know that:

$$\left\|\left[\nabla^2 \operatorname{Re}(\hat{g}(x^*))\right]^{-1} \nabla \operatorname{Re}(\hat{g}(x^*))\right\| \leq \frac{\epsilon_1}{\lambda_{min}\left(\nabla^2 \hat{f}_{k_h}(x^*)\right) - \epsilon_2} \tag{63}$$

Define:

$$\alpha = \frac{\operatorname{Lip}\left(\nabla^2 \operatorname{Re}(\hat{g})\right) \epsilon_1}{\left(\lambda_{min}\left(\nabla^2 \hat{f}_{k_h}(x^*)\right) - \epsilon_2\right)^2} \tag{64}$$

If $\alpha < \frac{1}{2}$, Newton–Kantorovich theorem [60; 33] states that:

$$\hat{x}^* \in B\left(x^*, \frac{\epsilon_1}{\lambda_{min}\left(\nabla^2 \hat{f}_{k_h}(x^*)\right) - \epsilon_2}\right) \tag{65}$$

$B(x, r)$ means a closed ball centered at $x$ with radius $r$, and $\hat{x}^*$ is the only critical point in the region.

Overall, what this means is that for any critical point $x^*$ of $\hat{f}_{k_h}$, if we choose:

1. $\epsilon_2 \leq \lambda_{min}\left(\nabla^2 \hat{f}_{k_h}(x^*)\right)$

2. $\epsilon_1$ such that $\alpha \leq \frac{1}{2}$

3. $\left\|\nabla \operatorname{Re}\left(\hat{g}(x)\right) - \nabla f_{k_h}(x)\right\| \leq \epsilon_1$ and $\left\|\nabla^2 \operatorname{Re}\left(\hat{g}(x)\right) - \nabla^2 f_{k_h}(x)\right\|_F \leq \epsilon_2$ both holds with probability at least $1 - \delta/2$.

Then, with probability least $1 - \delta$, there is only one critical point $\hat{x}^*$ of $\operatorname{Re}(\hat{g})$ such that: $\hat{x}^* \in B\left(x^*, \frac{\epsilon_1}{\lambda_{min}\left(\nabla^2 \hat{f}_{k_h}(x^*)\right) - \epsilon_2}\right)$.

∎

## K    PROOF OF THEOREM 6

The convergence of zeroth-order gradient methods using two-point gradient estimation over nonconvex but $L$-smooth function is established by Nesterov and Spokoiny [58], who showed the method converges to approximate stationary points at a sublinear rate. Specifically, in our case, with decaying step-size, the average squared gradient norm satisfies:

$$\min_{0 \leq l < T} \mathbb{E}_{u \sim \mathcal{N}(0, \mathbf{I}_d)} \left\|\nabla \operatorname{Re}\left(\hat{g}\left(x^{(l)}\right)\right)\right\|^2 = \mathcal{O}\left(\frac{1}{\sqrt{T}} + \mu^2 d^2\right) \tag{66}$$

The second term comes from the fact that gradient estimation is biased (due to the smoothing parameter $\mu$) but close to the real gradient. We can further derive a bound to quantify the result of zero-th order gradient ascent over $\operatorname{Re}(\hat{g})$: Let $l^* \in \{0 \cdots T\}$ be the iteration index such that $\mathbb{E}_{u \sim \mathcal{N}(0, \mathbf{I}_d)} \left\|\nabla \operatorname{Re}\left(\hat{g}\left(x^{(l)}\right)\right)\right\|^2$ is the smallest. Since $\operatorname{Re}(\hat{g})$ is real analytical, it satisfies Łojasiewicz inequality around a local mode $\hat{x}^*$:

$$\operatorname{Re}\left(\hat{g}\left(\hat{x}^*\right)\right) - \operatorname{Re}\left(\hat{g}\left(x^{(l^*)}\right)\right) \leq \mathcal{O}\left(\left\|\nabla \operatorname{Re}\left(\hat{g}\left(x^{(l^*)}\right)\right)\right\|^{2/2\theta}\right) \tag{67}$$

Where $\theta$ is the Łojasiewicz exponent. Combine with the descent lemma:

$$\frac{1}{2 \operatorname{Lip}\left(\nabla \operatorname{Re}(\hat{g})\right)} \left\|\hat{x}^* - x^{(l^*)}\right\|^2 \leq \operatorname{Re}\left(\hat{g}\left(\hat{x}^*\right)\right) - \operatorname{Re}\left(\hat{g}\left(x^{(l^*)}\right)\right) \leq \mathcal{O}\left(\left\|\nabla \operatorname{Re}\left(\hat{g}\left(x^{(l^*)}\right)\right)\right\|^{2/2\theta}\right) \tag{68}$$

Taking the expectation on both sides yields:

$$\mathbb{E}_{u \sim \mathcal{N}(0, \mathbf{I}_d)} \left[\left\|\hat{x}^* - x^{(l^*)}\right\|^2\right] \leq \mathcal{O}\left(\left(\frac{1}{\sqrt{T}} + \mu^2 d^2\right)^{1/2\theta}\right) \tag{69}$$

∎

## L    PROOF OF EQUATION 13

*Proof.* Want to show that $\phi_{k_h}(x + x') = \frac{D}{\sqrt{D}}\phi_{k_h}(x) \otimes \phi_{k_h}(x') \quad \forall x, x' \in \mathbb{C}^d$. Recall that $\phi(.)$ is:

$$\phi_{k_h}(x) = \frac{1}{\sqrt{D}}\left[e^{j(\omega_1^T x)}e^{j(\omega_2^T x)}e^{j(\omega_3^T x)}\ldots e^{j(\omega_D^T x)}\right] \tag{70}$$

So:

$$
\begin{aligned}
\frac{D}{\sqrt{D}}\phi_{k_h}(x) \otimes \phi_{k_h}(x') &= \frac{D}{\sqrt{D}}\frac{1}{\sqrt{D}}\left[e^{j(\omega_1^T x)}e^{j(\omega_2^T x)}e^{j(\omega_3^T x)}\ldots e^{j(\omega_D^T x)}\right] \\
&\quad \otimes \frac{1}{\sqrt{D}}\left[e^{j(\omega_1^T x')}e^{j(\omega_2^T x')}e^{j(\omega_3^T x')}\ldots e^{j(\omega_D^T x')}\right] \\
&= \frac{D}{\sqrt{D}}\frac{1}{D}\left[e^{j(\omega_1^T x)}e^{j(\omega_1^T x')}\ldots e^{j(\omega_D^T x)}e^{j(\omega_D^T x')}\right] \\
&= \frac{D}{\sqrt{D}}\frac{1}{D}\left[e^{j(\omega_1^T(x+x'))}\ldots e^{j(\omega_D^T(x+x'))}\right] \\
&= \frac{1}{\sqrt{D}}\left[e^{j(\omega_1^T(x+x'))}\ldots e^{j(\omega_D^T(x+x'))}\right] \\
&= \phi_{k_h}(x + x')
\end{aligned}
\tag{71}
$$

∎

## M    ABLATION STUDY

### M.1    HYPERPARAMETER SENSITIVITY

Here, we provide a sensitivity analysis on important RFMS hyperparameters, including the mapped dimension($d$), number of iterations ($T$), kernel bandwidth($h$), learning rate($\eta$), and smoothing parameter($\mu$). Experimental results here are done on S4 dataset. Unless specified in the table below, we use the default $D = 500, T = 100, h = 0.2, \eta = 0.003, \mu = 5e - 4$.

| S4 dataset | $D = 50$ | $D = 100$ | $D = 200$ | $D = 300$ | $D = 500$ |
|---|---|---|---|---|---|
| NMI | 64 | 67 | 68 | 69 | 69 |
| Time(s) | 1.7 | 2.7 | 6.2 | 10.3 | 17.8 |

| S4 dataset | T = 10 | T = 50 | T = 100 | T = 150 | T = 200 |
|---|---|---|---|---|---|
| NMI | 0 | 35 | 70 | 70 | 71 |
| Time(s) | 2.1 | 8.6 | 17.4 | 25.8 | 33.7 |

| S4 dataset | $h = 0.05$ | $h = 0.1$ | $h = 0.2$ | $h = 0.4$ | $h = 0.8$ |
|---|---|---|---|---|---|
| NMI | 50 | 62 | 67 | 10 | 0 |

| S4 dataset | $\eta = 0.00075$ | $\eta = 0.0015$ | $\eta = 0.003$ | $\eta = 0.006$ | $\eta = 0.1$ |
|---|---|---|---|---|---|
| NMI | 49 | 54 | 69 | 71 | 28 |

| S4 dataset | $\mu = 5e - 6$ | $\mu = 5e - 5$ | $\mu = 5e - 4$ | $\mu = 5e - 3$ | $\mu = 1$ |
|---|---|---|---|---|---|
| NMI | 69 | 70 | 70 | 64 | 5 |

We introduce $\eta$ as a step-size parameter so the mean-shift update matches the familiar gradient ascent form. This also lets us control how far we move each iteration. In practice, the kernel affects the scale of the update through bandwidth, and $\eta$ provides a convenient way to absorb/adjust that step magnitude and to improve stability under approximation/zeroth-order noise.

Among the hyperparameters, only $D$ and $T$ will affect the runtime of the algorithm. As claimed in the paper, runtime scales linearly in both $T$ and $D$. Larger $T$ and $D$ will almost certainly produce better results, which aligns with Theorems 5 and 6 in the paper that quantify the closeness between the result returned by RFMS and actual modes of KDE. RFMS can be sensitive to $h, \eta, \mu$. Similar to many machine learning approaches, those hyperparameters need to be chosen empirically based on the on-hand data. The advantage of RFMS is that it allows for much faster hyperparameter-tuning due the the algorithm's efficiency.

## M.2 MONTE-CARLO(MC) VS. QUASI MONTE-CARLO(QMC)

Here, we add additional experiments to disentangle the effect of Monte-Carlo sampling (MC) and Quasi Monte-Carlo sampling (QMC) in RFMS at varying $D$, better NMI are highlighted:

| NMI ($D = 50$) | a1 | a2 | a3 | unbalance | s1 | s2 | s3 | s4 |
|---|---|---|---|---|---|---|---|---|
| MC | 0.83 | 0.86 | 0.85 | 0.98 | 0.92 | 0.86 | 0.67 | 0.65 |
| QMC | **0.85** | 0.86 | **0.86** | 0.98 | **0.94** | **0.87** | **0.71** | **0.66** |

| NMI ($D = 100$) | a1 | a2 | a3 | unbalance | s1 | s2 | s3 | s4 |
|---|---|---|---|---|---|---|---|---|
| MC | 0.87 | 0.87 | 0.89 | 0.98 | 0.96 | 0.89 | 0.7 | 0.64 |
| QMC | **0.89** | **0.89** | **0.90** | 0.98 | **0.96** | **0.91** | **0.74** | **0.68** |

| NMI ($D = 200$) | a1 | a2 | a3 | unbalance | s1 | s2 | s3 | s4 |
|---|---|---|---|---|---|---|---|---|
| MC | 0.89 | 0.91 | 0.92 | 0.98 | 0.97 | 0.91 | 0.74 | 0.63 |
| QMC | **0.92** | **0.93** | **0.94** | 0.98 | **0.98** | **0.93** | 0.74 | **0.68** |

| NMI ($D = 300$) | a1 | a2 | a3 | unbalance | s1 | s2 | s3 | s4 |
|---|---|---|---|---|---|---|---|---|
| MC | 0.92 | 0.94 | 0.95 | 0.98 | 0.98 | 0.92 | 0.75 | 0.68 |
| QMC | **0.95** | **0.97** | **0.96** | 0.98 | 0.98 | **0.93** | **0.76** | **0.69** |

At the same $D$, QMC shows a consistent improvement over MC sampling as expected.

## N ADDITIONAL DETAILS ON CLUSTERING EXPERIMENTS

All clustering experiments were run on an Intel i5-11400 CPU. All algorithms are implemented in Python and use the NumPy [30] library. We provide further details on the clustering experiments below. For the cluster assignment after mean-shift, we construct a graph over shifted data points where an edge exists between $x$ and $x'$ if $k_h(x, x') > 0.9$ and run the connected component algorithm.

### N.1 DIFFERENT MEAN-SHIFT ALGORITHMS

Gradient mean-shift(GMS):

$$x^{(l+1)} = x^{(l)} + \eta \frac{\nabla f_{k_h}(x^{(l)})}{f_{k_h}(x^{(l)})} \tag{72}$$

Fixed-point iteration mean-shift(MS):

$$x^{(l+1)} = \frac{\sum_{i=1}^{n} k_h \left( x_i - x^{(l)} \right) x_i}{\sum_{i=1}^{n} k_h \left( x_i - x^{(l)} \right)} \tag{73}$$

Blurring mean-shift(BMS):

$$x^{(l+1)} = \frac{\sum_{i=1}^{n} k_h \left( x_i^{(l)} - x^{(l)} \right) x_i^{(l)}}{\sum_{i=1}^{n} k_h \left( x_i^{(l)} - x^{(l)} \right)} \tag{74}$$

For MS++, we refer the readers to the original paper by Jiang et al. [38] for a more detailed discussion.

## N.2 DATASET DETAILS

|  | a1 | a2 | a3 | unbalance | s1 | s2 | s3 | s4 |
|---|---|---|---|---|---|---|---|---|
| $n$ | 3000 | 5250 | 7500 | 6500 | 5000 | 5000 | 5000 | 5000 |
| # Cluster | 20 | 35 | 50 | 8 | 15 | 15 | 15 | 15 |
| $d$ | 2 | 2 | 2 | 2 | 2 | 2 | 2 | 2 |

|  | WallRobot | UserKnowledge | WirelessLocalization |
|---|---|---|---|
| $n$ | 5456 | 403 | 2000 |
| # Cluster | 4 | 4 | 4 |
| $d$ | 4 | 5 | 7 |

Table 3: Dataset information for clustering experiments.

## N.3 HYPERPARAMETER INFORMATION

We fix the number of iterations across all methods. Parameters such as $h$, $\mu$, $\eta$ are chosen impiricially. For RFMS, $D$ is chosen to be as small as possible while not degrading the result significantly.

|  | a1 | a2 | a3 | unbalance | s1 | s2 | s3 | s4 |
|---|---|---|---|---|---|---|---|---|
| Random Feature dimension $D$ | 300 | 300 | 300 | 300 | 300 | 300 | 300 | 300 |
| Bandwidth $h$ | 0.1 | 0.1 | 0.1 | 0.5 | 0.2 | 0.2 | 0.2 | 0.2 |
| Iteration $T$ | 100 | 100 | 100 | 100 | 100 | 100 | 100 | 100 |
| Smoothing parameter $\mu$ | 5e-4 | 5e-4 | 5e-4 | 5e-4 | 5e-4 | 5e-4 | 5e-4 | 5e-4 |
| Learning rate $\eta$ | 1e-3 | 1e-3 | 1e-3 | 1e-3 | 3e-3 | 3e-3 | 3e-3 | 3e-3 |

|  | WallRobot | UserKnowledge | WirelessLocalization |
|---|---|---|---|
| Random Feature dimension $D$ | 500 | 2000 | 750 |
| Bandwidth $h$ | 0.2 | 0.5 | 0.7 |
| Iteration $T$ | 100 | 100 | 100 |
| Smoothing parameter $\mu$ | 5e-4 | 5e-4 | 5e-4 |
| Learning rate $\eta$ | 1e-2 | 3e-2 | 5e-2 |

Table 4: RFMS hyperparameters.

|  | a1 | a2 | a3 | unbalance | s1 | s2 | s3 | s4 |
|---|---|---|---|---|---|---|---|---|
| Bandwidth | 0.15 | 0.15 | 0.15 | 0.3 | 0.15 | 0.15 | 0.15 | 0.15 |

|  | WallRobot | UserKnowledge | WirelessLocalization |
|---|---|---|---|
| Bandwidth | 0.2 | 0.5 | 0.5 |

Table 5: MS++ hyperparameter.

For MS, GMS, and BMS, we use the same hyperparameter whenever applicable.

### N.4 ADDITIONAL EXPERIMENTAL RESULTS

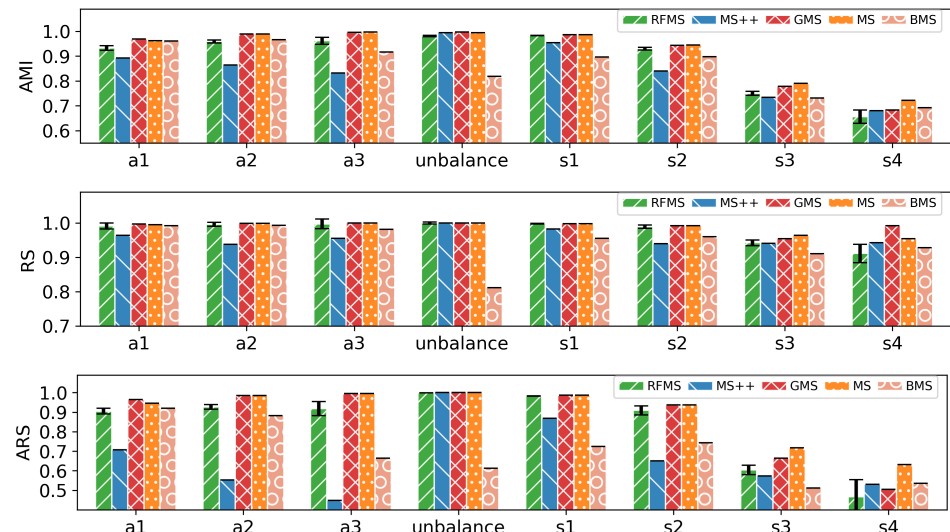

Figure 8: Adjusted mutual information(AMI), Rand Score(RS) and Adjusted Rand Score(ARS)

| Metric | Method | a1 | a2 | a3 | unbalance | s1 | s2 | s3 | s4 |
|---|---|---|---|---|---|---|---|---|---|
| AMI | RFMS | 0.93 ± 0.01 | 0.96 ± 0.01 | 0.96 ± 0.01 | 0.98 ± 0.00 | 0.98 ± 0.00 | 0.93 ± 0.01 | 0.75 ± 0.01 | 0.66 ± 0.03 |
| | GMS | 0.97 ± 0.00 | 0.99 ± 0.00 | 1.00 ± 0.00 | 1.00 ± 0.00 | 0.99 ± 0.00 | 0.94 ± 0.00 | 0.78 ± 0.00 | 0.68 ± 0.00 |
| | MS | 0.96 ± 0.00 | 0.99 ± 0.00 | 1.00 ± 0.00 | 1.00 ± 0.00 | 0.99 ± 0.00 | 0.95 ± 0.00 | 0.79 ± 0.00 | 0.72 ± 0.00 |
| | BMS | 0.96 ± 0.00 | 0.97 ± 0.00 | 0.92 ± 0.00 | 0.82 ± 0.00 | 0.90 ± 0.00 | 0.90 ± 0.00 | 0.73 ± 0.00 | 0.69 ± 0.00 |
| | MS++ | 0.89 ± 0.00 | 0.86 ± 0.00 | 0.83 ± 0.00 | 0.99 ± 0.00 | 0.95 ± 0.00 | 0.84 ± 0.00 | 0.73 ± 0.00 | 0.68 ± 0.00 |
| NMI | RFMS | 0.94 ± 0.01 | 0.96 ± 0.01 | 0.96 ± 0.01 | 0.98 ± 0.00 | 0.98 ± 0.00 | 0.93 ± 0.01 | 0.75 ± 0.01 | 0.67 ± 0.03 |
| | GMS | 0.97 ± 0.00 | 0.99 ± 0.00 | 1.00 ± 0.00 | 1.00 ± 0.00 | 0.99 ± 0.00 | 0.95 ± 0.00 | 0.78 ± 0.00 | 0.69 ± 0.00 |
| | MS | 0.96 ± 0.00 | 0.99 ± 0.00 | 1.00 ± 0.00 | 1.00 ± 0.00 | 0.99 ± 0.00 | 0.95 ± 0.00 | 0.79 ± 0.00 | 0.72 ± 0.00 |
| | BMS | 0.96 ± 0.00 | 0.97 ± 0.00 | 0.92 ± 0.00 | 0.82 ± 0.00 | 0.90 ± 0.00 | 0.90 ± 0.00 | 0.73 ± 0.00 | 0.69 ± 0.00 |
| | MS++ | 0.89 ± 0.00 | 0.87 ± 0.00 | 0.84 ± 0.00 | 0.99 ± 0.00 | 0.95 ± 0.00 | 0.84 ± 0.00 | 0.74 ± 0.00 | 0.69 ± 0.00 |
| ARS | RFMS | 0.91 ± 0.02 | 0.93 ± 0.01 | 0.92 ± 0.04 | 1.00 ± 0.00 | 0.98 ± 0.01 | 0.91 ± 0.02 | 0.60 ± 0.02 | 0.47 ± 0.09 |
| | GMS | 0.96 ± 0.00 | 0.99 ± 0.00 | 1.00 ± 0.00 | 1.00 ± 0.00 | 0.99 ± 0.00 | 0.94 ± 0.00 | 0.67 ± 0.00 | 0.51 ± 0.00 |
| | MS | 0.95 ± 0.00 | 0.99 ± 0.00 | 1.00 ± 0.00 | 1.00 ± 0.00 | 0.99 ± 0.00 | 0.94 ± 0.00 | 0.71 ± 0.00 | 0.63 ± 0.00 |
| | BMS | 0.92 ± 0.00 | 0.88 ± 0.00 | 0.66 ± 0.00 | 0.61 ± 0.00 | 0.72 ± 0.00 | 0.74 ± 0.00 | 0.51 ± 0.00 | 0.54 ± 0.00 |
| | MS++ | 0.71 ± 0.00 | 0.55 ± 0.00 | 0.45 ± 0.00 | 1.00 ± 0.00 | 0.87 ± 0.00 | 0.65 ± 0.00 | 0.57 ± 0.00 | 0.53 ± 0.00 |
| RS | RFMS | 0.99 ± 0.00 | 1.00 ± 0.00 | 1.00 ± 0.00 | 1.00 ± 0.00 | 1.00 ± 0.00 | 0.99 ± 0.00 | 0.94 ± 0.01 | 0.91 ± 0.03 |
| | GMS | 1.00 ± 0.00 | 1.00 ± 0.00 | 1.00 ± 0.00 | 1.00 ± 0.00 | 1.00 ± 0.00 | 0.99 ± 0.00 | 0.95 ± 0.00 | 0.92 ± 0.00 |
| | MS | 1.00 ± 0.00 | 1.00 ± 0.00 | 1.00 ± 0.00 | 1.00 ± 0.00 | 1.00 ± 0.00 | 0.99 ± 0.00 | 0.96 ± 0.00 | 0.95 ± 0.00 |
| | BMS | 0.99 ± 0.00 | 0.99 ± 0.00 | 0.98 ± 0.00 | 0.81 ± 0.00 | 0.96 ± 0.00 | 0.96 ± 0.00 | 0.91 ± 0.00 | 0.93 ± 0.00 |
| | MS++ | 0.96 ± 0.00 | 0.96 ± 0.00 | 0.96 ± 0.00 | 1.00 ± 0.00 | 0.98 ± 0.00 | 0.94 ± 0.00 | 0.94 ± 0.00 | 0.94 ± 0.00 |
| Time(s) | RFMS | 4.44 ± 0.05 | 8.36 ± 0.07 | 12.71 ± 0.17 | 10.63 ± 0.12 | 7.80 ± 0.14 | 7.71 ± 0.12 | 8.03 ± 0.10 | 7.93 ± 0.15 |
| | GMS | 30.24 ± 0.54 | 88.36 ± 4.21 | 175.98 ± 8.26 | 136.82 ± 2.19 | 75.12 ± 0.29 | 75.42 ± 0.47 | 76.82 ± 4.22 | 79.19 ± 3.96 |
| | MS | 16.98 ± 0.24 | 50.31 ± 2.04 | 93.03 ± 1.65 | 68.54 ± 0.36 | 45.21 ± 0.93 | 45.63 ± 0.74 | 45.80 ± 1.46 | 44.96 ± 0.34 |
| | BMS | 17.28 ± 0.24 | 47.16 ± 0.29 | 84.12 ± 0.51 | 68.34 ± 0.09 | 42.9 ± 0.24 | 43.21 ± 0.05 | 43.56 ± 0.04 | 43.37 ± 0.12 |
| | MS++ | 1.48 ± 0.05 | 2.70 ± 0.05 | 4.17 ± 0.12 | 4.10 ± 0.09 | 2.54 ± 0.02 | 2.63 ± 0.07 | 2.61 ± 0.07 | 2.61 ± 0.04 |

Table 6: Full clustering experimental results in table format under different metrics.

# O  ADDITIONAL DETAILS ON IMAGE SEGMENTATION EXPERIMENTS

Segmentation experiments were run on an Intel i5-11400 CPU. RFMS, MS, BMS, and MS++ are implemented in Python using the Numpy [30] library. QuickShift implementation comes from the Scikit-Image library [68]. Since RFMS, MS, and BMS might produce noisy clustering, we divide LAB color space into hypercubes of side length 100 after shifting for clustering assignment. For RFMS, we use $D = 10$, $\mu = 0.1$ and $\eta = 30$. RFMS, MS, and BMS use $h = 25$ whereas bandwidth for MS++ and Quickshift is 30 and 20, respectively.

## O.1  EXAMPLE SEGMENTATION RESULTS

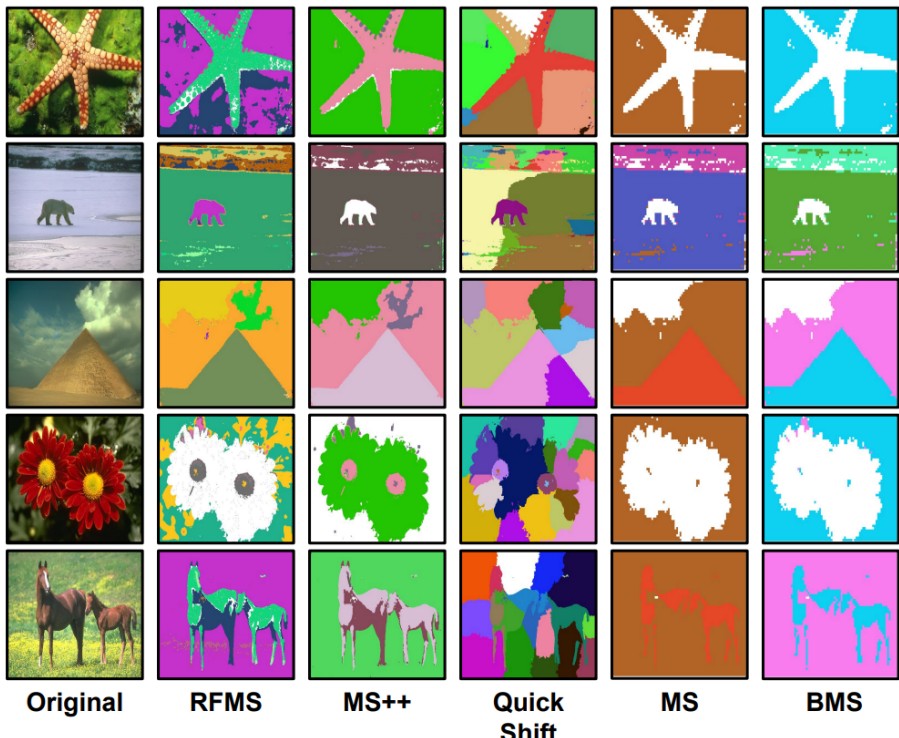

Figure 9: Visualization of segmentation result using different mean-shift based algorithms on BSDS500 dataset. Pixels belonging to the same cluster are marked with the same color.

## O.2  ADDITIONAL EXPERIMENTAL RESULT

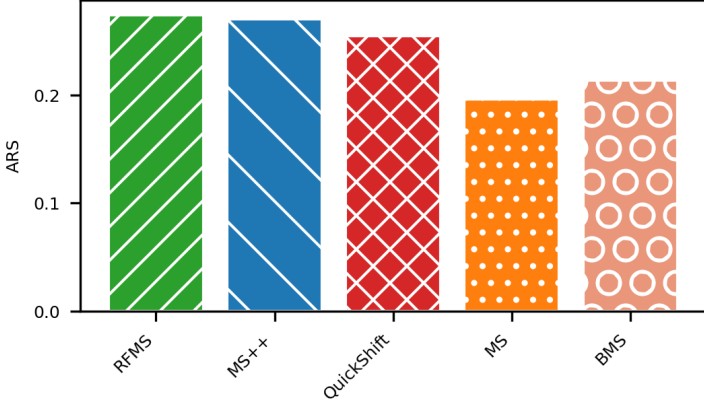

Figure 10: Adjusted Rand Score(ARS)

| FM (Fowlkes-Mallows score) | | | | |
|---|---|---|---|---|
| RFMS | MS++ | Quickshift | MS | BMS |
| 0.54 | 0.56 | 0.42 | 0.54 | 0.54 |

| ARS (Adjusted Rand Index) | | | | |
|---|---|---|---|---|
| RFMS | MS++ | Quickshift | MS | BMS |
| 0.27 | 0.27 | 0.25 | 0.19 | 0.21 |

| Time(s) | | | | |
|---|---|---|---|---|
| RFMS | MS++ | Quickshift | MS | BMS |
| 5000.11 | 5111.94 | 10557.48 | 15703.66 | 15019.48 |

Table 7: Full image segmentation experimental results in table format under different metrics.

## P   REPRODUCIBILITY / CODE AVAILABILITY

We value the availability and reproducibility of our work. The code and all the hyperparameters used in the experiment section are supplied as part of the supplemental material. We will also make our code publicly available upon acceptance of the paper.

## Q   LLM USAGE

Large Language Models (LLMs) were used during the preparation of this paper for assistance. Usage includes grammar, phrasing correction, polishing writing, and searching for or discovering related papers. All ideation, algorithms, technical novelties, and details are done by the authors. All LLM outputs were carefully reviewed and validated before inclusion in the manuscript.

