# OpenReview forum: "Random Feature Mean-Shift"
_ICLR.cc/2026/Conference — Submitted to ICLR 2026_

### Official Review · Reviewer_kFhF · 2025-10-24

**Soundness:** 3
**Presentation:** 3
**Contribution:** 3
**Rating:** 6
**Confidence:** 4

**Summary:**

The paper introduces a scalable, approximate version of the classical mean shift algorithm. This is based on approximating the Gaussian kernel utilised in kernel density estimators by means of Random Fourier Features (RFF). A theoretical analysis shows that, in the limit, the approximation recovers the modes of the "true" density. Experiments on clustering tasks show that the approach is faster and more effective than alternative fast mean shift variants in some cases.

**Strengths:**

* The authors introduce a simple approximate version of mean shift based on random Fourier features. Surprisingly, I could not find prior work that attempted this approach before (but I might have missed something).

* The authors attempt to justify their approximation theoretically, showing that, in the limit, it should recover the same grouping as the original mean shift algorithm.

* Experiments show empirically that the algorithm is competitive against other recent fast mean shift variants.

**Weaknesses:**

* I am not sure why the zero-th order approximation of the density gradient is needed. It should not be hard to take the analytical derivative of Eq. (2) w.r.t. $x$, nor should it be expensive to compute.

* The theorems in Sect. 5 show that the models of the original PDF map to modes of the approximated PDF, which is fine. However, they do not show that the approximated PDF does not introduce *additional* modes that could incorrectly trap the mean shift iteration, thus changing the resulting data clusters. In practice, I would expect that an approximation based on a truncated sinusoidal basis would (aka the RFFs) introduce additional modes with high likelihood.

* The paper would benefit from additional toy visualisation. Instead of only considering the final clustering results in Figure 4, it would be valuable to show the actual approximation of the PDF (in 1D and 2D). This would have the added benefit of illustrating additional unwanted local modes in the approximation, if any (although in practice these might be much more likely in higher dimensions). Likewise, particularly in a 2D case, it could be possible to visualise the original Comaniciu iteration vs the one implied by the approximation for a few selected points. This would also be rather illustrative.

* Mean shift has never been a very good algorithm for high-dimensional data clustering, and this method seems to further exacerbate the problem. Specifically, in Table 1, it seems that, for a large number of dimensions, the *original* mean shift is both more accurate and faster in two out of three cases.

Minor issues:

Mean shift was quite relevant in applications several years ago, but now it is much less so. This will somewhat limit the practical impact of this contribution.

There is likely a normalisation factor missing to the right of Eq. (2).

**Questions:**

* The authors invoke the "modern fixed point interaction" of mean shift [14] (line 126). I gather that, in their notation (eq. 1), this is obtained when $\eta = 1/(h^2cn)$. Given that the fixed-point iteration is guaranteed to converge, why should we choose anything but this value for $\eta$?

* Can the authors explain why one needs a randomised numerical approximation of the gradient instead of just computing it analytically (see also above)?

* Can the authors comment on the possibility that the approximated PDF has additional modes, and how likely these are to capture clusters in an unwanted manner?

* Can the authors discuss whether this technique is, in fact, useful with high-dimensional data (compared to just using the original mean shift)?

---

> ### Author Response · Authors · 2025-11-26
> **Response**
>
> Thank you for the constructive feedback! And thanks for recognizing our novelty of using RFF with mode-seeking; we believe it is a salient combination and can be a powerful primitive for future research. Please find below our responses to the weaknesses and questions:
>
> **W1. Why zeroth-order gradient estimation is needed instead of getting the gradient analytically over the RRF form of KDE?**
>
> The main reason is that an analytic gradient of equation 2 would require computing the Jacobian w.r.t every input dimension, which will result in an overall complexity of $O(Dd)$ instead of what we have now ($O(D)$), a zeroth-order gradient method. One major difference between RFMS and the grid-based mode-seeking approach (MS++) is that our complexity does not explicitly rely on the ambient $d$. Practically, what it means is that zeroth-order gradient estimation allows us to be a constant factor faster than calculating the RFF gradient analytically, with the constant factor depending on the ambient dimension of the data. Additionally, The Zeroth-order gradient approach only assumes we can evaluate density, not that the feature map is differentiable. This makes the algorithm applicable to other random feature maps where an analytic gradient is less convenient. We will include this discussion in a revised version of the paper to motivate the use of the zeroth-order gradient method.
>
>
> **W2. Could the RFF approximation introduce extra spurious modes that trap the mode-seeking process?**
>
> Regarding the possibility that the RFF KDE introduces additional modes: our mode-stability theorem already guarantees that every non-degenerate mode of the true KDE corresponds to a unique nearby mode of RRF KDE when the approximation is sufficiently accurate (uniform gradient/Hessian error, as in theorems 3 and 4), hence cluster structures are preserved up to small perturbations. Furthermore, if outside small neighborhoods of the true modes the KDE has gradient magnitude bounded below by $\gamma>0$, then whenever $\sup _x\|\nabla \operatorname{RFFKDE} (x)-\nabla \operatorname{KDE}(x) \| \leq \epsilon<\gamma$, RFF-KDE has no critical points outside those neighborhoods, and therefore cannot create spurious modes. Practically, any additional stationary points can only occur in very flat/low-density regions where $\|\nabla \operatorname{KDE}\|$ is tiny; these have negligible mode-seeking impact. In other words, intuitively, RFF approximation error behaves like small oscillatory noise; however, creating a new mode requires the gradient to flip sign and hit zero, meaning the perturbation must be comparable to the true gradient magnitude in that region, which empirically tends to be low-density regions and has minimal clustering impact. We will incorporate this discussion into the paper revision.
>
> **W3. Additional visualisations on 1D/2D examples.**
>
> We appreciate the suggestion that it would make the method more intuitive. We are currently working on those examples (RRF KDE surface vs. actual KDE surface) and will add them to the revised PDF here on OpenReview.
>
>
> **W4. RFMS in higher dimensions.**
>
> We fully agree that mode-seeking (mean-shift and KDE in general) suffers from the curse of dimensionality. Our goal here is not to fix this statistical issue, but to provide a computationally scalable approximation to classical mean-shift in regimes where it is still used. The large $d$ cases in Table 1 have relatively small $n$. In this regime, the $O(n^2)$ cost of classical mean shift is not yet prohibitive, so the complexity advantage of RFMS is not apparent. RFMS, on the other hand, offers its strongest speedups for large $n$ settings, which is the regime we emphasize in the main text and in the BSDS500 experiments.

---

> ### Author Response · Authors · 2025-11-26
> **Response Cont'd.**
>
> **Minor issue 1. Mean shift relevance**
>
> Despite its classical nature, mean-shift remains a fundamental and influential algorithm in many areas of machine learning. Its nonparametric, geometry-driven nature makes it a widely used primitive (even in recent years and at high-impact publication venues) for tasks such as self-supervised learning, image restoration, and outlier detection [1,2,3]. We provide a novel and useful way to resolve the longstanding bottleneck of quadratic runtime, which makes meaningful advancements over other methods with similar goals. Given the modern relevance of mean-shift, we believe that this principled advancement makes the paper appropriate for a high-impact venue like ICLR.
>
> **Minor issue 2. Normalization factor in Eq. (2)**
>
> We thank the reviewer for noticing this. It has been fixed and will be shown in a revised draft.
>
>
> **Q1. Choice of step size $\eta$**
>
> We introduce $\eta$ as a step-size damping parameter so the mean-shift update matches the familiar gradient ascent form: $x^{t+1}=x^t+\eta\left(\operatorname{FixPointIter}\left(x^t\right)-x^t\right)$. This does not change the fixed points (modes still satisfy $\operatorname{FixPointIter}(x)=x$ ), but it lets us control how far we move each iteration. In practice, the kernel affects the scale of the update through bandwidth ($\frac{1}{h^2}$ term), and $\eta$ provides a convenient way to absorb/adjust that step magnitude and to improve stability under approximation/zeroth-order noise., When $\eta=\frac{1}{h^2}$, it recovers the standard fixed-point mean-shift iteration ($\frac{1}{nc}$ term is absorbed into $\Phi$).
>
>
>
>
> **Q2. Why not take RFF-KDE gradient analytically?**
>
> Please see our response to Weakness 1, the main reason is the complexity difference of $O(D)$ and $O(Dd)$ per iteration per point for gradient, and the general applicability of the zeroth order method to other random feature mappings whose analytical gradients can be hard to obtain.
>
>
> **Q3. Additionally mode of RFMS**
>
> Our gradient/Hessian closeness guarantees (Theorems 3,4) imply that true modes remain and move only slightly. Extra modes, if any, must correspond to small oscillations. We will add toy visualisations and a discussion of this phenomenon, and explicitly acknowledge this in the paper. Please see our response to Weakness 2 for more details.
>
> **Q4. Usefulness in high dimensions vs original mean shift**
>
> Regarding high-dimensional data: in the ultra-high-dimensional regime, both RFMS and classical mean shift (or any KDE-based method) are fundamentally limited by the curse of dimensionality, so we do not claim improvement there. In low-to-moderate dimensions where mean shift is commonly applied, RFMS matches the original mean shift’s quality, while providing substantial runtime gains in the large $n$ regime because it avoids $O(n^2)$ kernel evaluations.
>
>
> We hope this addresses the reviewer's concerns, and we will incorporate the above discussion and components into the revision of the paper. Thanks again for the suggestions!
>
>
>
> [1] "Mean shift for self-supervised learning." ICCV'21.
>
> [2] "Mean-shift outlier detection and filtering." Pattern Recognition'21
>
> [3] "MeanShift++: Extremely fast mode-seeking with applications to segmentation and object tracking." CVPR'21

---

### Official Review · Reviewer_xt4T · 2025-10-25

**Soundness:** 3
**Presentation:** 3
**Contribution:** 2
**Rating:** 2
**Confidence:** 5

**Summary:**

This paper considers the mode finding and mean-shift problems using a Gaussian kernel.
The main idea is to use Rahimi-Recht's Random Fourier Features to represent the n points in a D dimensional space so that the core kernel computation of for the mode finding and mean-shift can be computed in O(D) time per point instead of O(n) time.

The algorithm RFMS shows moderate speed-up over regular mean shift when the data set grows from about n=2000 to about n=5000, with D = 500, 750, or 2000 -- without much loss in accuracy empirically.

There are also theoretical bounds.  About half are standard implications of RFFs, which show that it gets additive error about eps = 1/sqrt{D}.
There are also some potentially more interesting bounds about convergence -- but it was written in a form that made it hard to understand how long the algorithm would need to run in order to achieve a desired error bound.

**Strengths:**

This is a nice perspective an a core problem in ML that I would appreciate to be better understood.

**Weaknesses:**

The main downside of this paper is that it explored one out of many possible ways to approximate a KDE.  There are variety of ways based on

 - coresets  e.g., https://arxiv.org/abs/1802.01751
 - LSH  e.g., https://arxiv.org/abs/1808.10530
 - data structures e.g., https://arxiv.org/abs/2107.02736

and some of these ideas have also been applied to mode finding  https://arxiv.org/abs/1912.07673

What is notable about a number of these approaches (as demonstrated on the mode finding approach) is that
  - with more careful analysis, then one can find a *relative* error approximation, not the weaker additive approximation that this paper finds.
  - it is possible to approximate the KDE using roughly 1/eps time per query.  This would be roughly equivalent to needing sqrt(D) dimensions instead of the D required by RFF -- although the mechanism is different.

These settings are slightly different, so it is not immediately clear that they would directly achieve similar sorts of results.  But I suspect at least some of them are possible, and with improved bounds.  The main problem is that there is not a comparison.



The experimental section uses a decent number of datasets.  But the choice of D (the critical dimension parameter for RFFs) is chosen differently for just about each experiment without a lot of explanation.  This makes it so a potential user of this does not have much guidance on how to tune this parameter.  An explanation of this, or a fixed recommended choice, along with an ablation study is needed.

**Questions:**

Can you say anything formal or empirical about how this approach would compare to potential approaches based on coresets, LSH, or other data structures?

---

> ### Author Response · Authors · 2025-11-27
> **Response**
>
> Thank you for the careful review and for highlighting other alternatives for approximating KDE. Please consider the following point-by-point responses:
>
>
>
> **W1. One of many ways for approximating KDE:**
>
> We agree that KDE can be accelerated in several ways (coresets, LSH, or specialized data structures [1,3,4,5]). Our focus is specifically on an end-to-end mode-seeking algorithm whose per-iteration cost is linear in $n$ without discretizing the space or requiring specialized search structures that can degrade in higher dimensions (for example, kd-tree or other space partitioning
> structures). Review’s cited papers largely target fast queries of KDE at a point, but do not, to our knowledge, provide a principled, end-to-end mode-seeking procedure with guarantees on how modes of the approximate KDE move relative to the actual KDE modes (Theorems 3–5) and the convergence of a stochastic gradient-ascent style iteration to a neighborhood of those modes (Theorem 6). Propagating these KDE approximation errors to mode-seeking correctness is a unique contribution in our work.
>
> We also note that previous fast KDE methods likely will not work with the blurring version of mean-shift, as shifted points can require the reconstruction of the coreset or space partitioning structure every iteration. Whereas in RFMS, we only need to sum the updated point encoding, which is cheap and fast.
>
> We agree that RFMS can be viewed as one RFF-driven method in the broader design space of fast KDE approximation among coresets, LSH, and space partitioning data structures. Our goal is not to claim exclusivity, but to add a complementary member to this family: an approximation based on random Fourier features, which is attractive because it is simple, highly parallelizable, and comes with a well-established theoretical framework.
>
> We want to especially thank the reviewer for bringing those papers to our attention. In the revision, we will surely add a related works section, citing and discussing differences between our work and previous fast KDE approaches.
>
>
>
>
>
> **W2. Regarding Lee et al. “Finding the Mode of a KDE”[5]:**
>
> We thank the reviewer for highlighting the paper by Lee et al. [5]. While both papers study KDE's mode, they target different tasks and have different guarantee/computation characteristics, so they are best viewed as complementary within the KDE literature.
>
> First, Lee et al. focus on finding one global maxima of the Gaussian KDE, whereas RFMS is designed to run a mean-shift–style mode-seeking process for many modes, which is necessary in clustering and segmentation.
>
> Second, Lee et al.’s algorithm reduces the problem to solving systems of polynomial inequalities (with additional dimensionality reduction and recovery step for high-dimensional data). These are significantly more involved computations in practice than RFMS’s simple full vectorized computation (essentially just dot products and some simple element-wise operation). Notably, Lee et al. explicitly use a mean-shift step as part of their high-dimensional recovery procedure, underscoring the complementary nature of our work.

---

> ### Author Response · Authors · 2025-11-27
> **Response Cont'd.**
>
> **W3. Relative error vs additive error**
>
>
> We agree that relative-error guarantees are attractive, but for KDE, a uniform relative bound over all $x \in \mathbb{R}^d$ is generally ill-posed because the density can be arbitrarily close to 0 in low-density regions. Meaning the bound of form $\frac{|\operatorname{ApproxKDE}(x)-\operatorname{KDE}(x)|}{\operatorname{KDE(x)}}$ is unstable unless one restricts to regions with meaningful density values. This is why several fast KDE results are stated in additive form (coreset KDE, FGT [1,2] KDE approximation).
>
>
> Similarly, LSH and data-structure approaches [3,4] typically report relative error only for queries above a minimum density value. As an example, DEANN [4] treats the minimum KDE value $\tau$ as an explicit parameter when tuning estimators (Similarly for the LSH-based method [3], which also has a $\tau$ parameter in its KDE problem definition), reflecting the need to exclude near-zero regions for meaningful relative guarantees. Importantly, the mode-finding paper [5]  also makes this explicit assumption that $\max _x G_P(x) \geq \rho$, with their runtime bounds also dependent on $\rho$. Those indicate that a relative bound inherently requires a density lower bound.
>
>
>
>
> In our work, we therefore start from an additive RFF KDE bound uniformly over the entire space (also with a uniform bound for gradient and Hessian over the entire space). We would also like to note that for a subset $S$ in $\mathbb{R}^d$ such that $S=\left [x: \operatorname{KDE}(x) \geq \tau\right]$, our uniform additive error bound immediately implies a relative bound: $ \sup_{x\in S} \left|\frac{ \operatorname{RFF-KDE}(x) - \operatorname{KDE}(x)}{\operatorname{KDE}(x)} \right| \leq \frac{\varepsilon}{\tau}$. The nominator is exactly Theorem 2 in the paper.
>
>
>
>
>
>
>
> Moreover, mean-shift/mode-seeking does not just need KDE bounds. It needs the geometry of the KDE to be accurate (specifically, the gradients/Hessians near critical points, because modes are defined by zero gradient and local curvature). Uniform additive error is the natural notion of geometry in KDE for mode seeking because it is exactly what's needed for propagated KDE error to gradient/Hessian control, which yields stability bounds for not just one but all modes (Theorem 5). The contribution here is complementary and distinctive to previous work [1,2,3,4,5].
>
>
>
>
> **W4. Clarification on dimension $D$ and how long to run**
>
> We agree that $D$ (number of random features) and $T$ (number of iterations) are the main hyperparameters that control RFMS runtime. Our ablation (Appendix L) shows that the runtime grows linearly with both $D$ and $T$. Other RFMS hyperparameters have a negligible impact on runtime. In experiments, we keep $T$ fixed across methods, and pick $D$ to be the smallest value that does not noticeably hurt accuracy, reporting the chosen $D$.
>
> More broadly, we emphasize that data-driven tuning of hyperparameters that decides runtime and accuracy is unavoidable across fast KDE/mode-finding methods, because many quantities appearing in theory and describing the dataset’s geometry are unknown in practice.
>
> Other fast-KDE and approximate mode-finding methods [1,2,3,4,5] also require dataset-dependent hyperparameters (for example, hash function, data structure parameters, and minimal density value) but generally do not provide a clear, universal rule for choosing them because the needed geometric/density quantities are unknown a priori. In practice, these parameters are tuned empirically.
>
> Our advantage here is that RFMS is cheap to fine-tune because each trial run is linear-time (in n and D, per iteration), so sweeping $(D, T)$ to hit a desired runtime/quality point is fast compared with approaches that require heavier preprocessing (for example building complex data structures) or tuning parameters tied to an unknown density function.

---

> ### Author Response · Authors · 2025-11-27
> **Response Cont'd.**
>
> **Q1. Comparison with coresets, LSH, and other data structures**
>
>  Please see our response for weaknesses 1,2, and 3 for a more detailed discussion on our work in comparison with other fast KDE and the KDE mode finding paper, both in theory and computation.
>
>  In summary, the main difference from the cited fast-KDE [1,3,4] work is that we provide an end-to-end theoretical framework for mode seeking, not just a fast estimator for KDE values at query points. In particular, our guarantees propagate approximation error through the mode-seeking objective to obtain mode stability guarantees, whereas prior fast-KDE methods largely stop at KDE approximation. Compared to Lee et al. [5], we aim to recover all modes and the full modes structure, not only an approximate global maximum. Finally, computationally, RFMS uses only simple, highly-parallel operations, and its per-iteration cost scales well within the ambient dimension in the typical mean-shift regime (avoiding data-structure overhead and remaining efficient as $d$ grows).
>
> We hope this addresses the reviewer's concerns. In the revision, we will clearly cite and discuss differences between our work and previous literature for fast KDE and KDE mode-finding as we described above. We will also add discussion and derivation of the relative error bound for Theorem 2. Thanks again for the suggestions!
>
>
>
> [1] "Near-optimal coresets of kernel density estimates." Discrete \& Computational Geometry 2020
>
> [2] "Improved fast gauss transform and efficient kernel density estimation." ICCV'03
>
> [3] "Hashing-based-estimators for kernel density in high dimensions." FOCS'17
>
> [4] "Deann: Speeding up kernel-density estimation using approximate nearest neighbor search." PMLR'22
>
> [5] "Finding the mode of a kernel density estimate." ArXiv'19

---

### Official Review · Reviewer_m4YN · 2025-10-30

**Soundness:** 2
**Presentation:** 3
**Contribution:** 2
**Rating:** 4
**Confidence:** 3

**Summary:**

This paper proposes a new algorithm named Random Feature Mean-Shift (RFMS) for mode-seeking. This paper solved the poor scalability of the traditional mean-shift algorithm. RFMS successfully reduces the computational complexity from $O(n^2)$ to $O(n)$.

**Strengths:**

1. The paper demonstrates a sophisticated method that unites kernel approximation techniques with derivative-free optimization methods to solve a fundamental problem.
2. The framework delivers a complete theoretical structure which ensures algorithm reliability through its core approximation quality and iterate convergence properties.

**Weaknesses:**

1. The method produces kernel values that match the actual values within an error margin, which decreases at a rate of $O(1/\sqrt{D})$ and depends on both the number of dimensions and the kernel length-scale. The computation of gradients and derivatives, which mean-shift depends on, becomes more complex because it needs additional components or more stringent conditions. The convergence of your iterates to biased stationary points becomes more likely when the dimensionality is not sufficiently large.
2. The methods for small/medium dimension sizes produce deterministic acceleration results through IFGT and dual-tree FGT and grid-based MeanShift++, which provide precise control over errors at high speeds. The methods operate at linear-time complexity per iteration during practice without adding stochastic feature noise to the system. Thus, this method may underperform specialized fast KDE/mean-shift engines in low-d.
3. The combination of Low-discrepancy (Quasi Monte-Carlo) sampling with orthogonal/structured features decreases both variance and constants, but the approximation maintains its D-dependent asymptotic behavior. The approximation method faces difficulties when using small kernels because narrow bandwidths create challenges.

**Questions:**

See weakness.

---

> ### Author Response · Authors · 2025-11-26
> **Response**
>
> Thank you for the detailed feedback and for recognizing the value of combining random-feature kernel approximation with zeroth-order optimization for scalable mode-seeking. Please consider the following responses:
>
> **W1. RFF trade-off:**
>
> We agree that the central trade-off we inherit in RFMS is the same one faced by essentially all RFF methods: the approximation error decreases at the standard Monte-Carlo rate (improving accuracy requires increasing the number of random features $D$) [1,2]. However, this is a known, widely accepted trade-off in RFF literature because it replaces kernel computation with simple inner products in a feature space, yielding asymptotically faster algorithms and especially beneficial in the large-$n$ regime [3,4]. In our setting, the motivation is the same: when $n$ is large, the practical bottleneck of mean shift is the $O(n^2)$ pairwise kernel evaluation, and moving to $O(nD)$ is a significant scalability win even when $D$ is moderately larger than $d$. Importantly, RFMS does not impose more stringent modeling assumptions than standard RFF or mean-shift: the representation remains real-valued features with simple dot products - all operations would be required for standard mean-shift. We agree with the reviewer that the primary overhead is choosing $D$ to be big enough, but this cost becomes negligible once $n$ is the dominating cost (usually the case for mode-seeking).
>
>
> On the reviewer’s concern about “biased stationary points”, we would like to clarify that our KDE, gradient, and Hessian approximation bounds are all unbiased (theorems 1-4). The only potential bias comes from the standard smoothing in two-point zeroth-order gradient estimation, controlled by the smoothing parameter. Our Theorem 6 makes this dependence explicit and shows the bias term is tunable (decreasing with smaller $\mu$ and more iterations), and our analysis directly recommends using Theorems 5 and 6 together to quantify how close the RFMS output is to a true KDE mode. This is precisely why we included both results: Theorem 5 controls mode shift due to RFF, while Theorem 6 controls optimization error due to zeroth-order updates.
>
> Finally, we note that the same RFF trade-off has proven worthwhile in multiple large-scale applications where the goal is to replace quadratic kernel/attention computations with linear-time feature inner products[5]. Our work here follows the same motivation - using RFF in mode-seeking tasks to achieve a beneficial trade-off for a more efficient algorithm.
>
>
> **W2. Deterministic methods such as IFGT/FGT and MS++**
>
> Compared to IFGT/FGT and MeanShift++, RFMS achieves a genuinely simple linear complexity ($O(nD)$ per iteration) by replacing pairwise kernel evaluation with a single stored random-feature aggregate $\Phi$ and per-point inner products. While IFGT/FGT and MeanShift++ are also often described as linear, their speed relies on ambient dimension, along with bandwidth/accuracy-dependent constants (truncation or grid resolution), which can grow rapidly with $d$ (exponentially in the case of FGT and MS++, We shown that for MS++ in Table 1 of the paper). These specialized methods are excellent in very low-dimensional settings, but become less attractive as the ambient dimension grows. RFMS targets the complementary regime: large $n$ and low-to-moderate dimensions, where a stochastic, easily tunable approximation in $D$ is preferable. We thank the reviewer for bringing up additional deterministic mean-shift variants. We will include this discussion and comparison in the revision.
>
> Additionally, RFMS’s update loop only requires simple dot products and elementwise operations. By contrast, FGT/IFGT-style accelerations typically rely on nontrivial data structures or specialized hardware [6,7]. So while those methods can be excellent in low dimensions, RFMS offers a simpler drop-in implementation path: tune $D$ rather than engineering and tuning specialized data structures or dependencies.

---

> ### Author Response · Authors · 2025-11-26
> **Response Cont'd.**
>
> **W3: On QMC and narrow bandwidths:**
>
> The purpose of using QMC (and other improved random-feature constructions such as orthogonal/structured features) is primarily to improve efficiency in terms of the required number of features $D$ to achieve the same approximation quality with fewer features for lower runtime/memory. This is exactly how we position QMC in the paper, as shown in Figure 3. Achieving a universally faster asymptotic rate than $O(\frac{1}{\sqrt{D}})$ is generally unrealistic for random features methods without extra assumptions; this is why RFF papers typically focus on variance reduction rather than better asymptotic rates[8,9], which is what RFMS is designed to exploit here.
>
> On small bandwidths, we agree that narrow kernels are harder to approximate because the the spectral distribution; our uniform bound makes the dependency on $h$ explicit (Theorems 2-4). We also report RFMS sensitivity to $h$ empirically (Appendix L). The mitigating factor for RFMS is that increasing $D$ (or using QMC or other RFF variants) provides a direct way to push approximation error down, and the overall runtime remains linear in $n$ and tunable via $D,T$.
>
> We also note that the small bandwidth regime is challenging for any KDE/mean-shift method, not just RFMS. When $h$ becomes very small, the KDE landscape emerges with many sharp local maxima, which typically fragments clusters and makes mode-based clustering unstable. In this regime, even the exact mean shift is likely to converge to many tiny modes, and the resulting clustering is often not meaningful. Practically, mean shift is used with bandwidths that smooth at the scale of the intended clusters. As a result, while narrower kernels require more features $D$ to approximate (as reflected in our bounds), this generally does not limit RFMS in the typical clustering regime, and the feature budget $D$ remains a tunable knob to tighten approximation when needed.
>
>
> We hope this addresses the reviewer's concerns, and we will incorporate the above discussion and components into the revision of the paper. Thanks again for the suggestions!
>
>
> [1] "Random features for large-scale kernel machines." NeurIPS'07
>
> [2] "Optimal rates for random Fourier features." NeurIPS'05
>
> [3] "Random Feature Attention." ICLR'21
>
> [4] "Towards a unified analysis of random fourier features." JMLR'21
>
> [5] "Random features for kernel approximation: A survey on algorithms, theory, and beyond." TPAMI'21
>
> [6] "Fast \& accurate gaussian kernel density estimation." VIS'21
>
> [7] "Accelerating mean shift image segmentation with IFGT on massively parallel GPU." MIPRO'13
>
> [8]  "Orthogonal random features." NeurIPS'16
>
> [9] "Improved random features for dot product kernels." JMLR'24

---

### Official Review · Reviewer_7jCs · 2025-11-01

**Soundness:** 2
**Presentation:** 3
**Contribution:** 3
**Rating:** 6
**Confidence:** 2

**Summary:**

This paper introduces Random Feature Mean-Shift (RFMS), a scalable and theoretically grounded variant of the classical mean-shift algorithm. By employing random Fourier feature approximations and zeroth-order stochastic optimization, the authors reduce the computational complexity of mean-shift updates from quadratic to linear time. The proposed approach eliminates the need for expensive pairwise kernel evaluations and naturally extends to both standard and blurring mean-shift formulations.

**Strengths:**

The paper tackles the quadratic time complexity of the standard mean-shift algorithm by proposing a linear-time solution that leverages random feature approximation and zeroth-order optimization.

For the proposed RMFS method, the authors present comprehensive theoretical analyses, including concentration bounds and convergence guarantees.

Empirical evaluations demonstrate substantial efficiency improvements across multiple datasets while preserving accuracy comparable to existing approaches over clustering tasks and image segmentation.

**Weaknesses:**

The authors provide several theorems establishing upper bounds on the estimation errors. However, the sharpness or order of these bounds is not fully discussed.

\
It would be helpful if the authors could elaborate on how tight these bounds are, both in theory and in practice, and clarify whether they differ from or improve upon existing theoretical results in prior work.
A more detailed discussion on the asymptotic order and optimality of the bounds would strengthen the theoretical contribution.

**Questions:**

Refer to Weaknesses.

---

> ### Author Response · Authors · 2025-11-26
> **Response**
>
> We thank the reviewer for the thoughtful feedback and for recognizing the scalability and the theoretical depth of RFMS. We agree that discussing the order/tightness of the bounds more explicitly would strengthen the paper, and we will add this in the revision. Please consider the following responses:
>
> **1. Difference from prior works:**
>
> Prior RFF theory largely focuses on approximating kernel values or sometimes improving the uniform-error bound. In contrast, our theoretical contribution is an end-to-end framework for RFF-based mode-seeking:
>
> + We start from density approximation bounds (Theorems 1–2).
> + We then provide explicit gradient/Hessian approximation bounds (Theorems 3–4), which are the quantities underlying the geometry of the mode-seeking procedure.
> + Crucially, we turn these derivative bounds into a mod stability guarantee (Theorem 5): under standard non-degeneracy assumptions, each critical point of the true KDE has a nearby corresponding critical point for the approximated density, with a strong uniqueness statement in a neighborhood.
> + Finally, we connect the algorithmic procedure (two-point zeroth-order ascent) to known convergence guarantees for gradient-free optimization (Theorem 6), completing the chain from approximation $\rightarrow$ geometry of modes $\rightarrow$ iterates returned by RFMS.
>
> We believe this “kernel approximation + mode stability + zeroth-order optimization” synthesis is nontrivial and substantial, because it directly addresses what matters for mean-shift: not just approximating the kernel function, but preserving the mode structure that defines clusters.
>
> **2. Bound tightness:**
>
> Our Theorem 1 gives a pointwise concentration bound with the canonical Monte-Carlo/RFF rate. The estimation error decays on the order of $O(\frac{1}{\sqrt{D}})$. This $O(\frac{1}{\sqrt{D}})$ dependence is the standard (and, in general, optimal) rate for plain i.i.d. Monte-Carlo feature sampling, which is exactly the regime RFMS targets. Theorems 3 and 4 extend the approximation guarantees beyond function values to the gradient and Hessian. We view these as structurally tight in the sense that they retain the same Monte-Carlo dependence in $D$, and the additional factors are the usual price of controlling derivatives rather than kernel values. The primary contribution here is to provide a high-probability bound of KDE under RFF-induced perturbations, which is novel and necessary. In short, we agree the constants can be loose (Hoeffding/covering number arguments are conservative), but the rates in $D$ are optimal for vanilla i.d.d. Monte-Carlo RFF. And the $O(\frac{1}{\sqrt{D}})$
> error decay behavior is sufficient for the downstream mode-tracking results.
>
>
> **3. Tightness in practice:**
>
> In practice, our theory is reflected in a very direct accuracy–efficiency knob: since the RRF KDE is an unbiased estimator of the KDE w.r.t. the feature randomness, increasing the number of features $D$ reduces variance and improves the approximation almost surely. Empirically, we observe monotone behavior where larger $D$ and more iterations $T$ almost certainly produce better results while runtime scales linearly in $D$ and $T$, making tuning practical.
>
> This aligns with the uniform guarantee in Theorems 3 and 4. This is important because it means that for the tightness, one can make the approximation arbitrarily tight by choosing $D$ sufficiently large.
>
> **4. Significance:**
>
> We also want to emphasize that the theoretical component here is a meaningful and worthwhile contribution: rather than presenting only standard RFF concentration, we develop a complete pipeline tailored to mean-shift mode seeking. While Theorems 1 and 2 align in order with classical RFF results, the key novelty here is connecting random-feature approximation to preservation of modes and mean-shift dynamics, which (to our knowledge) is not addressed by prior RFF analyses that focus on kernel/value approximation alone.
>
> We hope this addresses the reviewer's concerns, and we will incorporate the above discussion and components into the revision of the paper. Thanks again for the suggestions!

---

### Author Response · Authors · 2025-12-04
**Rebuttal & revision summary**

Dear Area Chairs and Program Chairs,

Thank you for handling this year's ICLR reviewing process. We have revised the paper to directly address reviewers' concerns (marked in blue in the updated PDF file. Source files and supplementary materials have also been updated accordingly). Below we summarize the key changes and clarify why the updated manuscript resolves core feedbacks.

* Summary of revisions:

    1. We added a comprehensive discussion on fast KDE approximation approaches (coresets, LSH, special data structures, and fast Gauss transform) and mode-finding algorithms beyond mean-shift **(modification in section 2 related works and discussion C.1 in the appendix)**. We emphasize that the contribution of our work moves beyond just approximating KDE in a theoretically grounded way, but also how those approximation errors propagate into mode-seeking settings. Additionally, in contrast to other mode-finding algorithms, which can only find a global maximum with computation-heavy primitives, RFMS finds all modes with simple and efficient operations. This addresses the primary concerns raised by reviewers **xt4T**, **7jCs** and **m4YN**.

    2. We incorporate the usual minimal density value assumption and derive an additional relative bound for RFF-based density estimation **(modification in section 5 estimation and convergence bounds)**, effectively addressing reviewer **xt4T**'s comment on the bound-type.

    3. We added reasoning for using the zeroth-order gradient method instead of getting the gradient of the RFF-approximated KDE analytically **(modification in section 4 random feature mean-shift)**. The primary reason is a complexity gain from $O(Dd)$ to $O(D)$ per iteration per point. This addresses reviewer **kFhF**'s question.

    4. We add discussion, analysis, and additional experimental results on the quality of RFF-approximated KDE, and explain how it will not introduce spurious modes that trap the mode-seeking process **(Modification in section 5 on bounds and RFMS density estimation visualization in Appendix B)**. This addresses reviewer **kFhF** questions regarding spurious modes and the request for additional experiments and visualization of KDE-approximation quality.

    5. We highlighted the modern relevance of mean-shift algorithms and their limitations/potential extension of RFMS for higher-dimensional data **(Modification in D.3 and C.5 of Appendix)**. Despite the classical nature and known limitations of mean-shift, it remains the de facto algorithm for mode seeking and a powerful primitive. This answers the questions raised by reviewer **kFhF**.

    6. We provide additional context and interpretation on the tightness of bounds **(Modification in C.3 in Appendix)**. Showing that theorems 1-4 have a rate of $O(\frac{1}{\sqrt(D)})$, which are the key notions of tightness and are sufficient for our mode-tracking guarantees. This answers the questions raised by reviewer **7jCs**.

    7. We clarified the trade-off that all RFF-based methods inherit **(Modification in section 4 random feature mean-shift)**. Despite the approximation nature and dependency on $D$, RFF is widely accepted and adopted as it often leads to asymptotically better algorithms, which is the case here for our work. This answers the concerns raised by reviewer **m4YN**.

    8. We add small changes to the abstract, introduction, and conclusion sections to clarify the main contribution of our work: an end-to-end and theoretically grounded fast mode-seeking method beyond the usual kernel approximation literature. We also add small changes such as missing normalizing constant (**kFhF**) and the choice of hyperparameters (**kFhF, xt4T**) in various places throughout the paper.

We did not receive reviewer responses due to the reviewer information leak incident. Nevertheless, we have revised the manuscript to address the concerns and questions raised in the initial reviews to the best extent possible. We respectfully request that the program committee consider these additions and clarifications in the final decision.

Best regards,

Submission 12056 Authors

---

### Meta-Review · Area_Chair_sYzt · 2025-12-16

**Summary:**

This paper offers a clean random-feature-based view of mean-shift with a largely complete theoretical pipeline and some empirical speedups. However, two reviewers are clearly negative and one is only mildly positive. Core concerns are not yet addressed: most guarantees resemble standard RFF theory, with limited demonstration that they yield practically stronger or more informative behavior than existing fast KDE/mode-finding methods; there are no direct comparisons to strong deterministic alternatives; and empirical evidence in difficult regimes is not sufficient. Even after rebuttal, novelty and external validity are below the acceptance bar of the competitive ICLR, and thus I recommend reject.

**Reviewer Concerns:**

**7jCs** Concerns: sharpness/tightness of the bounds; how these differ from standard RFF results; limited guidance on when the guarantees are practically informative. Addressed: discussions on the rates, tightness, and interpretation of Theorems 1 to 4. Outstanding: bounds still largely mirror classical RFF behavior and provide limited concrete guidance on regimes where RFMS is clearly preferable in practice.

**m4YN** Concerns: RFF error scaling with dimension/bandwidth; risk of biased stationary points; potential underperformance versus baselines in low dimensions; difficulty with narrow kernels. Addressed: clarified the RFF trade-offs and intended regimes of advantage, and discussed QMC/structured features. Outstanding: no direct empirical comparisons to strong deterministic baselines; limited evidence in challenging bandwidth regimes.

**xt4T** Concerns: limited engagement with broader fast-KDE/mode-finding literature; weaker additive bounds; ad-hoc choice of feature dimension. Addressed: expanded related work; relative-error style bound under density assumptions; some guidance on dimension/hyperparameters. Outstanding: relative guarantees and hyperparameter selection remain weakly substantiated.

**kFhF** Concerns: need for zeroth-order gradients; possible spurious modes from RFF; lack of visualizations; weak performance/high-dimension relevance. Addressed: justified zeroth-order complexity benefits; added analyses/visualizations of KDE quality and spurious modes. Outstanding: quantitative evidence on high-dimensional behavior and robustness to extra modes.

**Reviewer Scores:**

**7jCs** Likely keeps a weak accept 6.

**m4YN** Likely stays below accept 4.

**xt4T** Likely remains strong reject 2 (may also raise to 4).

**kFhF** Likely keeps a weak accept 6.

---

### Decision · Program_Chairs · 2026-01-26

Reject